# Dissecting heterogeneity in malignant pleural mesothelioma through histo-molecular gradients for clinical applications

Yuna Blum [1], Clément Meiller[2,3], Lisa Quetel[2,3], Nabila Elarouci[1], Mira Ayadi [1], Danisa Tashtanbaeva[2,3], Lucile Armenoult[1], François Montagne[2,3,4,5,12], Robin Tranchant [2,3,13], Annie Renier[2,3], Leanne de Koning[6], Marie-Christine Copin[5,7], Paul Hofman[8,9], Véronique Hofman[8,9], Henri Porte[4,5], Françoise Le Pimpec-Barthes[2,3,10,11], Jessica Zucman-Rossi [2,3], Marie-Claude Jaurand[2,3], Aurélien de Reyniès[1] & Didier Jean [2,3]

Malignant pleural mesothelioma (MPM) is recognized as heterogeneous based both on histology and molecular profiling. Histology addresses inter-tumor and intra-tumor heterogeneity in MPM and describes three major types: epithelioid, sarcomatoid and biphasic, a combination of the former two types. Molecular profiling studies have not addressed intra-tumor heterogeneity in MPM to date. Here, we use a deconvolution approach and show that molecular gradients shed new light on the intra-tumor heterogeneity of MPM, leading to a reconsideration of MPM molecular classifications. We show that each tumor can be decomposed as a combination of epithelioid-like and sarcomatoid-like components whose proportions are highly associated with the prognosis. Moreover, we show that this more subtle way of characterizing MPM heterogeneity provides a better understanding of the underlying oncogenic pathways and the related epigenetic regulation and immune and stromal contexts. We discuss the implications of these findings for guiding therapeutic strategies, particularly immunotherapies and targeted therapies.

[1] Programme Cartes d'Identité des Tumeurs (CIT), Ligue Nationale Contre Le Cancer, 75013 Paris, France. [2] Centre de Recherche des Cordeliers, Sorbonne Universités, Inserm, UMRS-1138, 75006 Paris, France. [3] Functional Genomics of Solid Tumors, USPC, Université Paris Descartes, Université Paris Diderot, Université Paris 13, Labex Immuno-Oncology, 75000 Paris, France. [4] Service de Chirurgie Thoracique, Hôpital Calmette - CHRU de Lille, 59000 Lille, France. [5] Université de Lille, 59045 Lille, France. [6] Translational Research Department, Institut Curie, PSL Research University, 75005 Paris, France. [7] Institut de Pathologie, Centre de Biologie-Pathologie, CHRU de Lille, 59037 Lille, France. [8] Laboratoire de Pathologie Clinique et Expérimentale (LPCE) et biobanque (BB-0033-00025), CHRU de Nice, 06003 Nice, France. [9] Université Côte d'Azur, 06108 Nice, France. [10] Assistance Publique-Hôpitaux de Paris, Hôpital Européen Georges Pompidou, 75015 Paris, France. [11] Département de Chirurgie Thoracique, Hôpital Européen Georges Pompidou, 75015 Paris, France. [12] Present address: Service de Chirurgie Générale et Thoracique, CHU de Rouen, 76000 Rouen, France. [13] Present address: Laboratoire de Biochimie (LBC), ESPCI Paris, PSL Research University, CNRS UMR8231 Chimie Biologie Innovation, 75005 Paris, France. Correspondence and requests for materials should be addressed to A.d.R. (email: aurelien.dereynies@ligue-cancer.net) or to D.J. (email: didier.jean@inserm.fr)

**M**alignant pleural mesothelioma (MPM) is a rare and very deadly cancer. Current therapeutic options are curative in a very small fraction of patients. As for most cancer types, clinical trials have highlighted MPM diversity in terms of the prognosis and patient response to anticancer agents[1], suggesting underlying tumor heterogeneity. Better understanding of tumor heterogeneity in its various dimensions is thus of paramount importance for the identification of therapeutic strategies leading to patient cures. Tumor heterogeneity is studied both at the inter-tumor and intra-tumor levels and considers not only tumor cells but also their microenvironments. Pathologists have described three major histological MPM types: epithelioid (MME), sarcomatoid (MMS), and biphasic (MMB), the latter of which consists of a mix of MME and MMS. Large-scale molecular profiling studies have also emphasized MPM heterogeneity[2,3], which may explain the difficulty in defining a unique and specific biomarker for MPM[4]. Asbestos, which is the main risk factor for MPM, may contribute to this molecular heterogeneity, because it causes a wide variety of molecular aberrations[5]. Several molecular stratifications of MPM have been proposed recently and are related to the histology and prognosis and partly to specific mutations[3,6–8].

According to histological observations, two contrasted tumor cell populations (epithelioid and sarcomatoid) are found in MPM. We apply a deconvolution approach to decompose each bulk MPM molecular profile as a combination of both populations. This novel approach quantifies tumor heterogeneity and avoids a strict subtype assignment based on subjective hierarchical classifications, which do not take into account intermediate phenotypes and show intrinsic limitations regarding intra-tumor heterogeneity. We also integrate epigenetic data to search for possible underlying regulatory mechanisms and to identify potential key regulators. This new way of thinking about the pathology is a step toward a better understanding of the biology underlying MPM heterogeneity. Moreover, it provides a significant contribution to clinical applications with implications for prognosis and therapeutic strategies, including immunotherapies and targeted therapies.

## Results

**Unsupervised clustering analyses reveal molecular gradients.** First, we performed unsupervised hierarchical clustering, using a consensus method based on 3 different linkages (ward, complete, and average) and bootstrap resampling, on our transcriptomic dataset we generated (Affymetrix array on 63 frozen MPM tumor samples) and identified two molecular subtypes of MPM (Supplementary Figure 1a). These subtypes were associated with the histology and prognosis and were independent of the sex, age, asbestos exposure or stage of the disease (Supplementary Figure 1a, c, Supplementary Figure 2). These results were consistent with our previous study based on cell line transcriptomic data, which showed two MPM subtypes with the same clinical characteristics, i.e., separation of histologic types and a worse prognosis for the C2 subtype[6]. Centroid-based prediction of the previous cell line-based subtypes using our frozen tissue series confirmed the consistency of both classifications (Supplementary Figure 1a, c). Then, we assessed intra-subtype heterogeneity by performing unsupervised clustering on the transcriptomic profiles of the C1 and C2 samples separately. Each subtype was subdivided into two groups (C1A and C1B, and C2A and C2B, respectively) (Supplementary Figure 1b, d, e). Subsequently, we compared these subtypes with those already published[3,6] and those defined by unsupervised clustering from other public series (i.e., the Gordon, Lopez and TCGA series)[9–11] (Supplementary Figure 3). We performed a meta-analysis to compare all of the

clusters from the different classifications by correlating the centroids of their corresponding meta-profiles (Fig. 1a). This analysis highlighted two main groups of highly correlated clusters present in all datasets that corresponded to the most extreme epithelioid and sarcomatoid phenotypes, which contained our C1A subtype and the *Epithelioid* subtype from the Bueno series[3] or our C2B subtype and the *Sarcomatoid* subtype from the Bueno series[3] respectively. Apart from these two opposite meta-clusters, the remaining clusters did not form robust meta-clusters shared across the different classification systems, suggesting that they might simply reflect various cut-offs of a continuum that combined epithelioid and sarcomatoid components. This important observation led us to rethink the pathology in terms of molecular gradients and to consider each sample as a mixture of these two components. We used weighted in silico pathology (WISP), a novel deconvolution method aiming at assessing intra-tumor heterogeneity by estimating the proportion of pure entities in bulk molecular profiles. WISP is a two-step approach that first estimates pure population profiles based on predefined pure samples and then estimates the proportion of these pure populations in a mixed sample based on the first step output (see Methods section for more details). We considered a MPM sample as a mixture in various proportion of epithelioid-like, sarcomatoid-like and non-tumor component. For a given sample the sum of these three proportions was therefore equal to 1. The first step of WISP was performed on representative samples for each component from our transcriptomic data (Affymetrix array on 63 tumor samples and 4 normal samples) that were selected as described in the Method section. The second step aiming at estimating the proportion of each component was applied on all available tumor tissue samples ($n = 442$) from our transcriptomic data and the different public transcriptomic datasets (Reynies[6], Gordon[10], Lopez[9], TCGA[11], and Bueno[3] series). We named these molecular components E-comp and S-comp to distinguish them from the epithelioid and sarcomatoid histologically defined morphologies and their proportions in a given sample as the *E*-score and *S*-score, respectively.

As shown in Fig. 1b, the *E*-score and *S*-score estimated in all available tumor tissue samples ($n = 442$) from the different transcriptomic datasets led to opposite gradients for E-comp and S-comp (Fig. 1b, Supplementary Figure 4a). Of note, 90% of the samples analyzed shows a tumor content based on WISP estimations greater than 75% tumor content, suggesting the presence of few tumor samples with low tumor content, which are found in all tumor series. These molecular gradients were related to the histology and the different molecular classifications predicted in the whole sample set of 442 samples (Fig. 1b–d, Supplementary Figure 4a, b).

To facilitate a clinical transfer of E-score and S-score estimations, we performed qRT-PCR analysis on the 63 tumors used in our Affymetrix dataset (CIT exploration series) and 114 new cases (CIT validation series) (Supplementary Data 1). We reproduced the results obtained from the Affymetrix data with a signature of 55 genes and a reduced signature of up to 15 genes (Supplementary Figure 5a, b, c). The *E*-score and *S*-score were predicted in the validation series, from which we could confirm their associations with the histology types (Supplementary Figure 5d).

**Biological characterization of the molecular components.** To gain biological mechanistic information for E-comp and S-comp, first we identified genes whose expression positively correlated with these components (Supplementary Data 2). As expected, 110 genes identified in other studies as overexpressed in MME or in MMS, were significantly positively correlated with the *E*-score or *S*-score. In particular *UPK3B*, *MSLN*, *CLDN15* were significantly

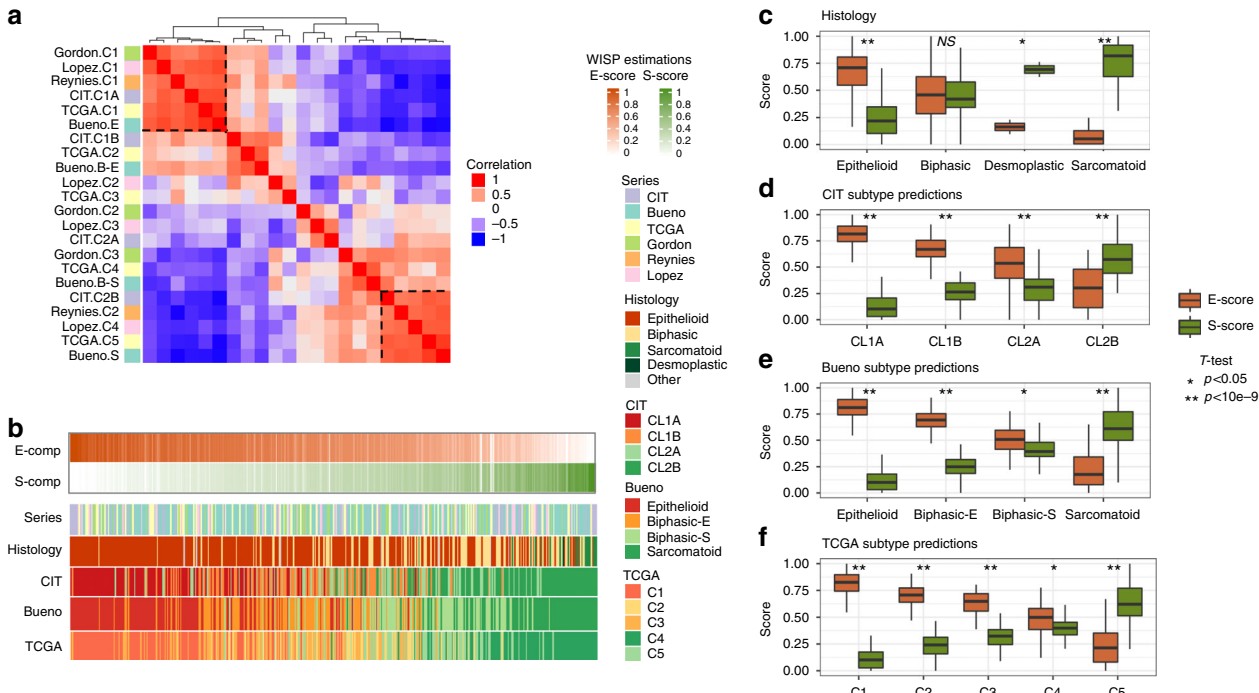

**Fig. 1** Meta-analysis and molecular gradients. **a** Correlation matrix of centroid profiles of all clusters from the different classifications. **b** Estimation of the *E*-score and *S*-score and classification subtype predictions in all available tumor tissue samples (442 samples). The samples were ordered based on their *E*-score and *S*-score ratios. **c–f** Boxplots of the *E*-score and *S*-score according to the histology results (**c**), CIT subtype predictions (**d**), Bueno subtype predictions (**e**), and TCGA subtype predictions (**f**). Significance in the *T*-test comparing the *E*-score and *S*-score in each modality is shown (*$P$ value < 0.05, **$P$ value < 10e−9, NS, not significant). For all boxplots (**c**, **d**, **e**, **f**), bottom and top of boxes are the first and third quartiles of the data, respectively, and whiskers represent the lowest (respectively highest) data point still within 1.5 interquartile range of the lower (respectively upper) quartile. Center line represents the median value

positively correlated to the *E*-score and *LOXL2* and *VIM* to the *S*-score[3,9]. Our study also highlighted new genes such as *PDZK1IP1* and *AXL* that were positively correlated to the *E*-score or *S*-score, respectively. *PDZK1IP1* encodes the MAP17 cargo protein; this protein plays an important role in diseases involving chronic inflammation, which is a characteristic of asbestos-related cancer[12]. *AXL* is a receptor tyrosine kinase and a member of the TAM (Tyro3, Axl, and Mer) family[13] (Supplementary Data 2). Then, we performed enrichment analysis of genes whose expression levels correlated with the *E*-score and/or *S*-score using KEGG, GO, and Reactome databases. We identified specific pathways associated with each component (Fig. 2a, Supplementary Figure 6, Supplementary Data 3). The EMT (Epithelial–mesenchymal transition), TP53 signaling, cell cycle, angiogenesis and immune checkpoints were positively associated with the *S*-score, whereas pathways involving cell junctions and complement and several metabolic pathways were positively associated with the *E*-score.

Then, we characterized E-comp and S-comp at the genetic level by focusing on the major altered genes (*CDKN2A*, *NF2*, *BAP1*, and *TP53*) during mesothelial carcinogenesis using available mutation and copy number alteration (CNA) data from the TCGA series. Significant positive associations were observed between *NF2* and *TP53* genetic alterations and the *S*-score (Supplementary Figure 7). Interestingly, higher occurrence of *TP53* mutations was reported in MPM tumors with a sarcomatoid contingent[3].

Next, we characterized E-comp and S-comp at the epigenetic level using methylome data (HumanMethylation450 Beadchip, 295009 CpG after preprocessing) and at the miRNA level using miRNome data (Illumina HiSeq 2000, 861 miRNA after preprocessing). These methylome and miRNome data were

generated from 62 and 60 MPM tumor samples included in our Affymetrix dataset (CIT exploration series), respectively. Several component-specific pathways showed a high proportion of genes whose expression levels correlated with the DNA methylation level, suggesting that methylation might play a major role in their regulation (Fig. 2b). We correlated the methylation levels of CpG sites with the *E*-score and *S*-score. Interestingly, CpG sites whose methylation level correlated with the *S*-score were preferentially located in CpG islands, in contrast to those whose methylation levels correlated with the *E*-score, which were mainly located in non-CPG islands (Supplementary Figure 8). Consistent with this result, we showed that the CpG island methylator phenotype (CIMP) index was positively correlated with the *S*-score as well as with *DNMT1* and *IDH2* expression, both of which are known to be associated with hypermethylation, especially in CpG islands (Supplementary Figure 9)[14,15]. To the best of our knowledge, only one study has reported differentially methylated genes between MME and MMS; a total of 17/72 of these genes were also identified by our analysis, 4 of which were associated with expression deregulation and were correlated with the molecular gradients (Supplementary Data 4)[16]. Surprisingly, among the genes known to be frequently hypermethylated in MPM compared to the normal pleura, we observed heterogeneity in their methylation profile that correlated with the *E*-score and *S*-score. Among the 502 genes identified by Christensen et al.[16], 48 and 25 were positively correlated with the *E*-score and *S*-score, respectively (Supplementary Data 4). Furthermore, other genes identified as hypermethylated in MPM in different studies showed methylation profiles that were positively correlated with the *S*-score (*ESR1*, *DAPK3*, *SYK*, and *TMEM30B*) or *E*-score (*FHIT* and *KAZALD1*)[17]. We also analyzed known oncogenes and tumor suppressors and observed a difference in their

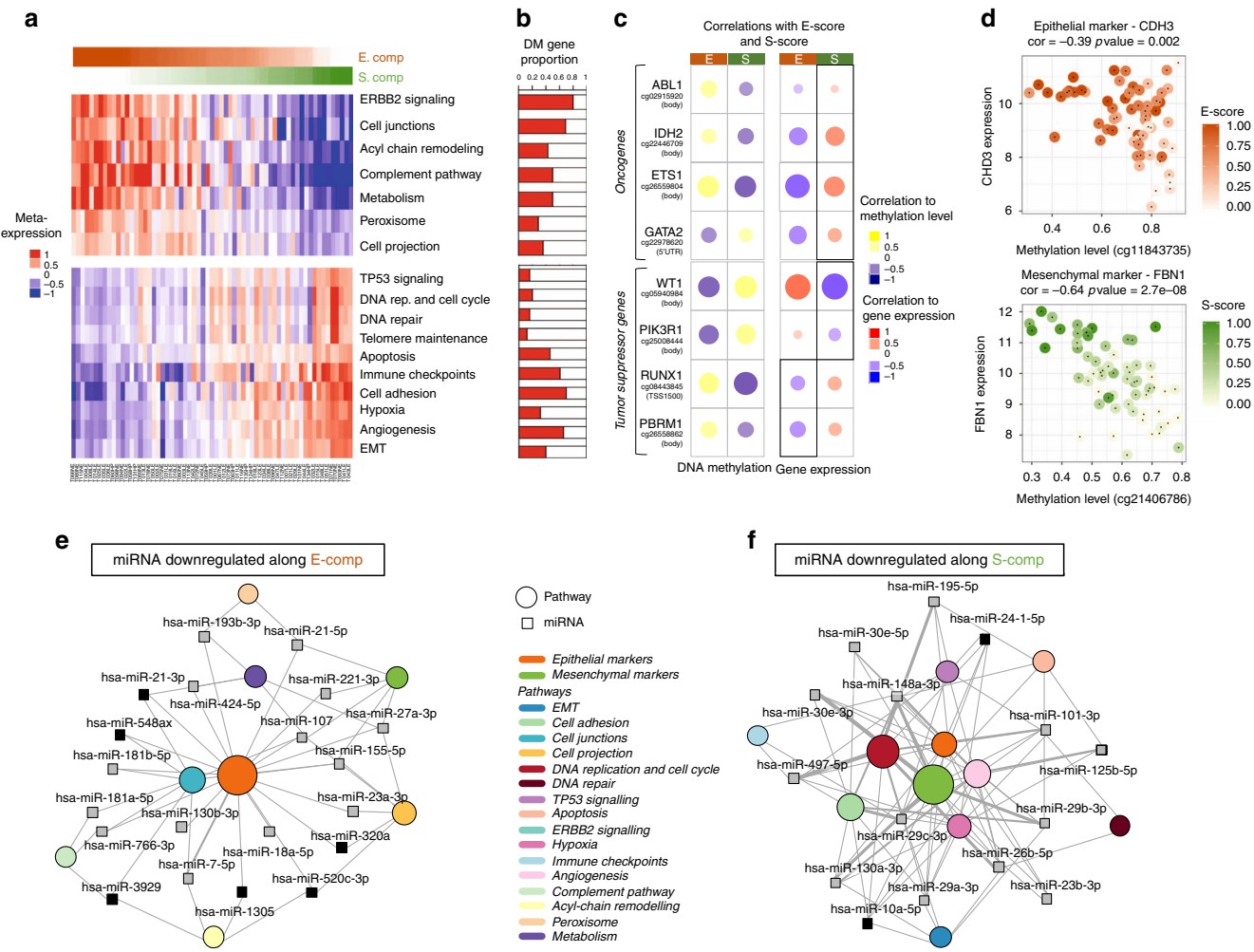

**Fig. 2** Component-specific pathways and epigenetic regulation. **a** Expression heatmap of the component-specific pathways (*P* value < 0.05, Fisher's exact test). For each pathway, the mean expression profile of the associated deregulated genes was calculated. Samples were ordered by the ratio of rescaled *E*-score and *S*-score. **b** Lateral bars correspond to each pathway and to the proportion of genes whose expression levels were correlated with the DNA methylation level (DM, differentially methylated). **c** Heatmap of the correlations between DNA methylation and gene expression with the *E*-score and *S*-score for known oncogenes and tumor suppressor genes (FDR adjusted *P* value < 0.05, Pearson's correlation test). The circle size is proportional to the correlation coefficient value. **d** Correlation plots between the DNA methylation level and the *CDH3* (epithelial marker) and *FBN1* (mesenchymal marker) gene expression levels; color gradient changes correspond to the *E*-score or *S*-score. For each plot, the correlation coefficient and *P* value are shown (Pearson's correlation test). **e, f** The network displays miRNAs whose expression levels are negatively correlated with the *E*-score (**e**) or *S*-score (**f**) and their targeted pathways (among the component-specific pathways between E-comp and S-comp) based on validated miRNA-target associations. The network was restricted to miRNAs that were negatively correlated with the expression of their target (FDR adjusted *P* value < 0.05, Pearson's correlation test) and targeted at least 2 (**e**) out of 5 (**f**) different component-specific pathways. The edge thickness is proportional to the number of genes in the pathway targeted by the miRNA. The circle size of each pathway is proportional to the number of genes targeted by the miRNA in the network. miRNAs that have been described in the MPM literature are represented by a gray square, and undescribed miRNAs are represented by a black square

methylation profiles that was dependent on the *E*-score and *S*-score. This result suggests that distinct (intra-cellular) oncogenic mechanisms are present in E-comp or S-comp related cells. (Fig. 2c). For instance, the tumor suppressor genes *WT1* and *PI3KR1* were hypermethylated and underexpressed in high *E*-score tumors whereas *RUNX1* and *PBRM1* were hypermethylated and underexpressed in high *S*-score tumors. The oncogenes *ETS1*, *ABL1*, and *IDH2* were hypomethylated and overexpressed along S-comp. We also identified known epithelial and mesenchymal markers[6] (53 and 24, respectively, Supplementary Data 4) with gradual methylation and expression along one of the components. For instance, the mesenchymal marker *FBN1* was hypomethylated and overexpressed along S-comp (Fig. 2d). Conversely, the epithelial marker *CDH3* was hypomethylated and overexpressed along E-comp (Fig. 2d).

In addition, we integrated our miRNA dataset, obtained from 60 MPM included in our Affymetrix dataset (CIT exploration series), and identified crucial miRNA-target regulation based on two validated miRNA-target association databases (miRTarbase and TarBase). As shown in Fig. 2e, d, the miRNA regulatory networks highlighted central miRNAs that targeted various component-specific pathways associated with increases in the *E*-score or *S*-score. We identified many miRNAs targeting epithelial or mesenchymal markers that were downregulated along the E-comp or S-comp, respectively. Among the hub miRNAs, miR-21-5p and miR-21-3p were underexpressed along E-comp and were associated with activation of different pathways, including cell junctions and peroxisome. Consistently, miR-21 was previously described in the literature as underexpressed in MME compared to the other histological types[18] and was

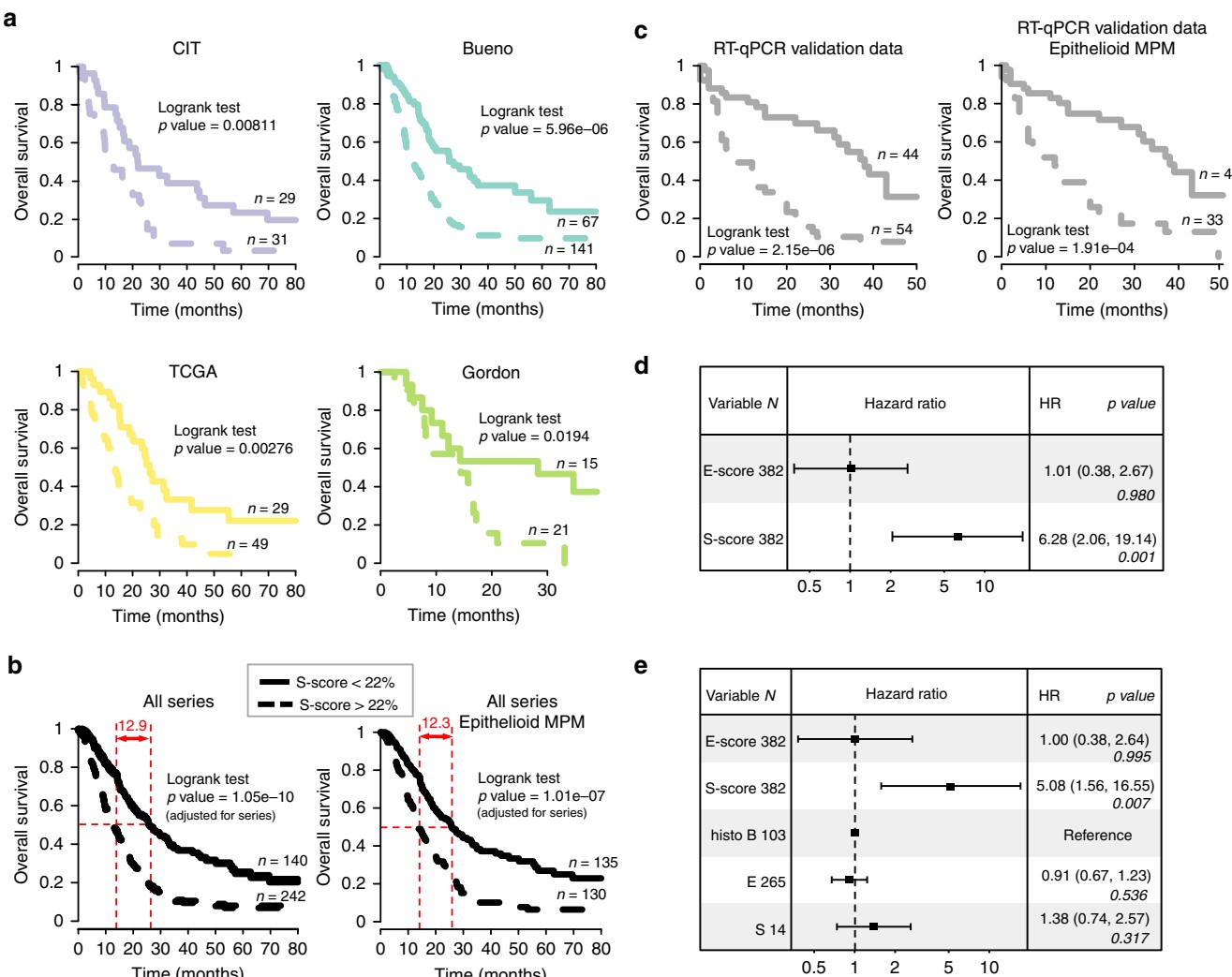

**Fig. 3** Prognostic impact of the *S*-score. Overall survival curve plots for patients with less than 22% of *S*-score (plain curve) or more than 22% of *S*-score (dashed curve) in samples in the CIT, Bueno, TCGA, and Gordon series (**a**), all samples from the different series or the series restricted to epithelioid MPM (**b**), and our qRT-PCR validation dataset (**c**). Difference in median overall survival between patients with less or more 22% of S-score is indicated in red (**b**). The *P* values are based on a log-rank test and adjusted for the series when appropriate, and the number of patients (*n*) is indicated in each condition. **d**, **e** Forest plots of the overall survival hazard ratios (HRs) estimated using a multivariate Cox analysis adjusted for the series, integrating the *E*-score, *S*-score (**d**) and the histological types (**e**) for MPM samples from the CIT, Bueno, TCGA and Gordon series. The hazard ratios, 95% confidence intervals (CIs) and related Wald test *P* values are given for both components

considered an oncogenic miRNA in several types of cancer[19]. In contrast, several central miRNAs were downregulated along S-comp, including miR-148a-3p, which is known to negatively regulate the EMT in lung cancer[20]. miR-148a-3p targets were implicated in several signaling pathways that were activated with S-score increases (Supplementary Figure 10). We also identified hub miRNAs that were previously described in MPM with similar expression profiles, including miR-101-3p, miR-195-5p and miR-29c-3p; a lower expression level of the latter predicted a worse prognosis[21].

Our analysis revealed other hub miRNAs that may contribute to pathway deregulation depending on the *E*-score and *S*-score that were not previously described in MPM but were implicated in other cancers, such as miR-10a-5p[22], miR-24-1-5p[23], and miR-520c-3p[24].

Finally, we identified miRNAs with strong positive associations with the *S*-score, such as miR-3929-3/5p and miR-1305, or with the *E*-score, such as miR-148a-5p, that could be interesting biomarkers for MMS and MME, respectively (Supplementary Data 5).

**Association with prognosis and drug sensitivity**. To investigate the clinical relevance of the MPM molecular gradients, we investigated their potential impacts on prognosis and drug sensitivity.

First, we analyzed the overall survival of patients in the different series according to their *E*-score and *S*-score. Interestingly, the presence of S-comp was associated with a worse outcome in each series (Fig. 3a) or when all series were analyzed together (Fig. 3b) using a robust cut-off of 22% of *S*-score determined after a bootstrap procedure testing thresholds ranging from 0.1 to 0.5 (Supplementary Figure 11), even when the series were restricted to histologically diagnosed epithelioid MPM (Fig. 3b, Supplementary Figure 12a, b). To validate our findings, we tested the same cut-off of 22% of *S*-score in our qRT-PCR validation dataset and confirmed the prognostic impact of the *S*-score (Fig. 3c). Then, we performed a survival meta-analysis on all available tumor samples by applying a bivariate Cox proportional hazards regression model adjusted by tumor series (Fig. 3d). Remarkably, the analysis showed a hazard ratio (HR) of 6.28 (*P* value = 0.001) for the *S*-score. Furthermore, a Cox model

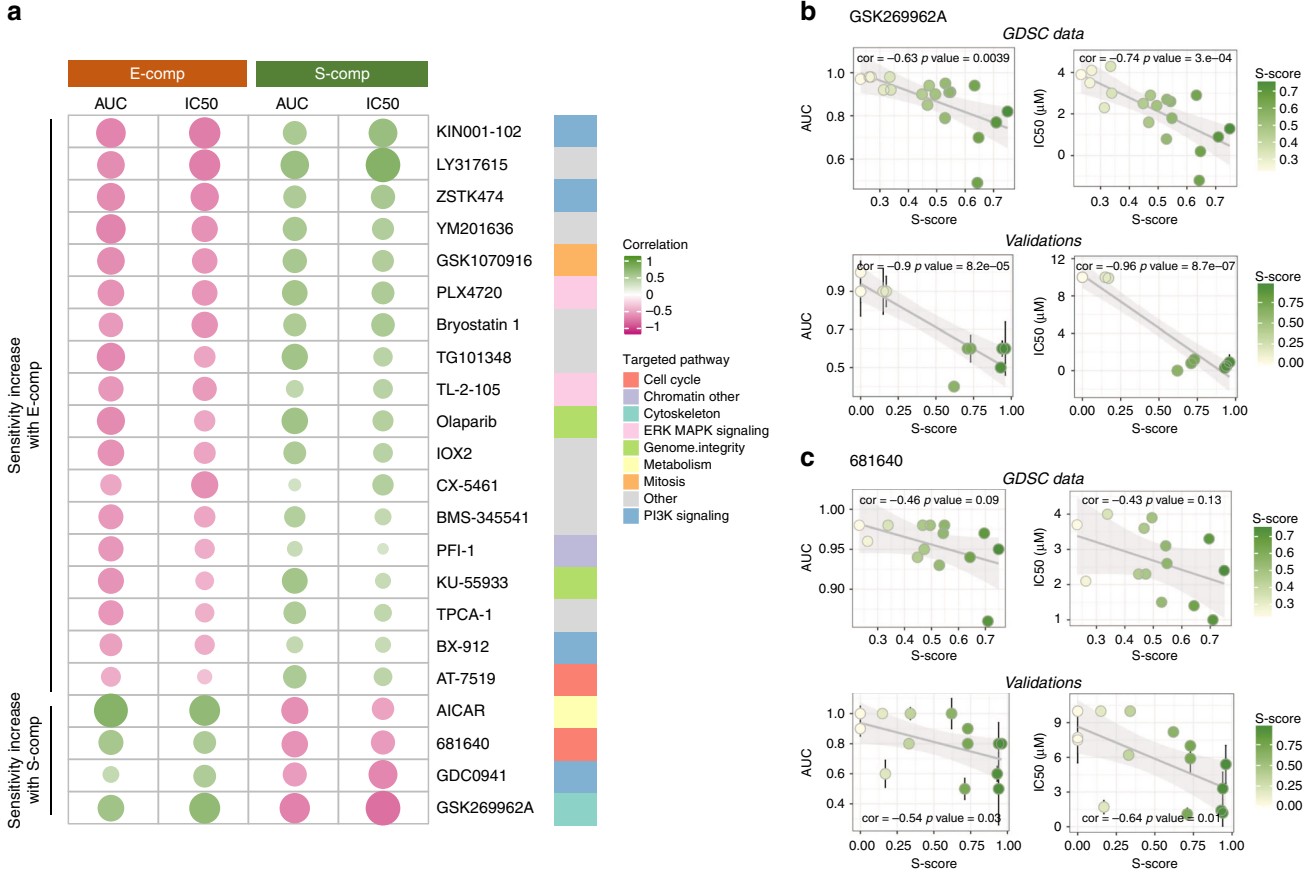

**Fig. 4** Association with drug sensitivity. **a** Heatmap of the correlations between the AUC and IC50 and *E*-score and *S*-score for the drugs included in the GDSC database with at least one significant correlation with the components (*P* value < 0.05). The circle size is proportional to the correlation value. The drugs are ordered on either side by their negative correlations with the *E*-score or *S*-score, which corresponded to a sensitivity increase along the component. The signaling pathways targeted by these inhibitors are mentioned on the right. **b, c** AUC and IC50 plots (error bars correspond to s.d.) obtained from the GDSC data or determined from our validation experiments vs. the *S*-score for GSK269962A (**b**) and 681640 (**c**). The color gradient represents the range of *S*-score. For each plot, the correlation coefficient and *P* value are shown (Pearson's correlation test)

integrating the *E*-score, *S*-score, and the histological types showed the superiority of the *S*-score for assessment of the MPM prognosis (Fig. 3d). Similar results were obtained when integrating pre-existing subtyping systems[3,6] into the Cox model, which strengthened the prognostic value of the molecular gradients (Supplementary Figure 12c, d). Significance of the *S*-score was also observed when performing a cox model restricted to patients showing high tumor content samples (combined *E*-score and *S*-score > 90%) (Supplementary Figure 12e).

To further assess the clinical relevance of the MPM molecular gradients, we analyzed the Genomics of Drug Sensitivity in Cancer (GDSC) database[25], in which drug responses were determined for various compounds on up to 21 MPM cell lines. A deconvolution signature defined using the transcriptomic data of the cell lines from our previous publication[6] was first used to estimate *E*-score and *S*-score in GDSC cell lines. Interestingly, the presence of both components in a given cell line supports that tumor heterogeneity was preserved in cell culture. Other cell types are also preserved in culture such as cancer stem cells (CSC). Published data support that CSC are still present in MPM cell lines[26]. We identified 22 compounds for which the cell response estimated by the AUC and IC50 correlated to the E-score and/or S-score (Fig. 4a, Supplementary Data 6).

Three compounds (the Akt1/2 kinase inhibitor KIN001-102, the Rho-associated protein kinase (ROCK) inhibitor GSK269962A, and the Wee1 inhibitor 681640 were selected

based on the correlation between the compound sensitivity (AUC and IC50) and the *E*-score or *S*-score (Fig. 4a). To validate the correlations between the inhibitor sensitivity and the *S*-score and *E*-score, cell viability was measured on 12–17 MPM cell lines in cultures treated with the three inhibitors. Normalized AUC and IC50 values were calculated from the cell viability curves (Supplementary. Figure 11, supplementary Data 7). The GSK269962A and 681640 inhibitors had strong effects on cell viability in some MPM cell lines in culture, in contrast to the effects of KIN001-102, which induced a slight decrease in viability at high concentrations in most of the MPM cultures. A significant positive correlation with the *S*-score was confirmed for the two former inhibitors (Fig. 4b, c) but not for KIN001-102 with the *E*-score (Supplementary. Figure 12).

**Molecular components are related to specific immune contexts.** Recent clinical trials have demonstrated major responses with anti-PDL-1 or anti-PD1 immunotherapies in many types of cancers, including MPM[27]. In this context, we analyzed immune populations using the Microenvironment Cell Population Counter (MCP-counter) tool on our transcriptomic dataset[28]. This tool aims to provide robust relative quantification of the abundance of immune and non-immune stromal cell populations from the transcriptome analysis of a heterogeneous sample. We validated the performance of MCP-counter in MPM by

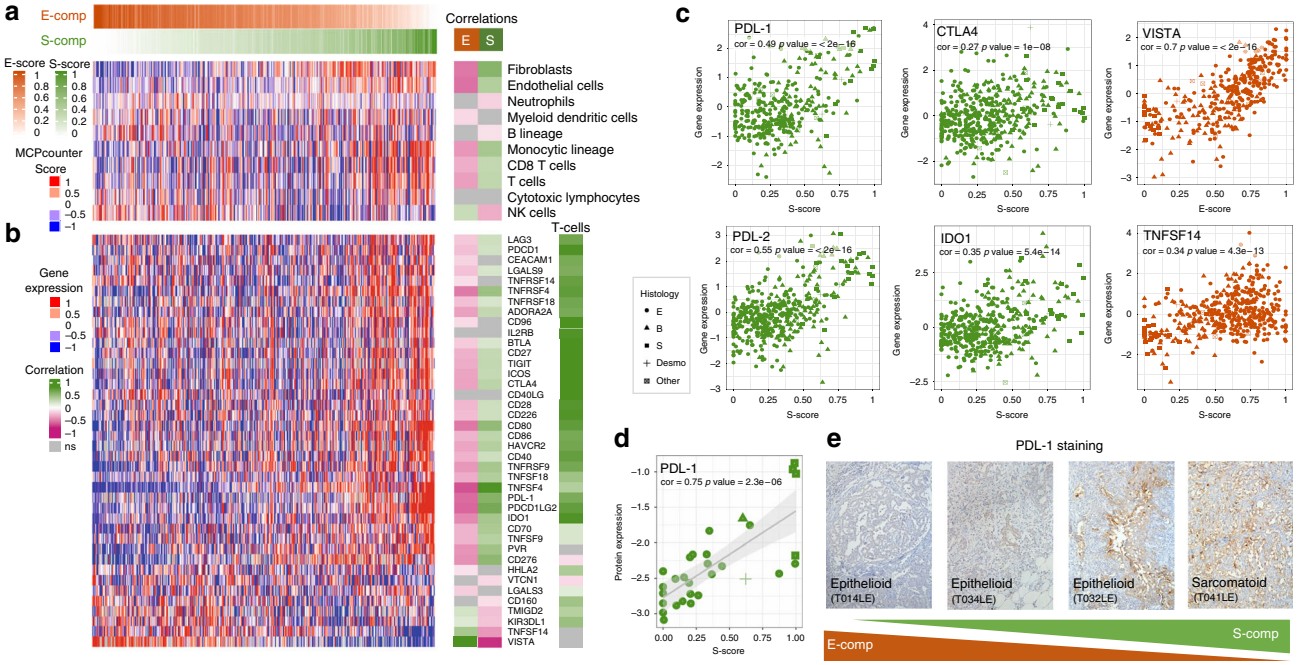

**Fig. 5** Specific immune landscape. **a**, **b** Heatmaps showing quantification of immune and stromal populations computed by MCP-counter from transcriptomic data (**a**) and immune checkpoint (ICK) gene expression (**b**), along E-comp and S-comp. The lateral red-green heatmap displays correlation coefficient values between the corresponding features and the E-score or S-score and between ICK and the T cell infiltration score. Non-significant correlations (P value > 0.05, Pearson's correlation test) are represented in gray. **c**, **d** Gene expression plots for different immune checkpoints (*PDL1*, *PDL2*, *CTLA4*, *TNFSF14*, and *VISTA*) and one checkpoint modulator (*IDO1*) (**c**) and the plots of PDL-1 protein expression (RPPA data) (**d**) vs. the *E*-score or *S*-score. For each plot, the correlation and P value are shown (Pearson's correlation test). The point shapes correspond to the different MPM histologies: epithelioid (E), biphasic (B), sarcomatoid (S), and desmoplastic (Desmo). **e** Immunohistochemical staining for PDL-1 in four samples along E-comp and S-comp

immunohistochemistry (IHC) on immune and stromal populations (Supplementary Figure 15). Correlation analyses between the *S*-score and MCP-counter estimations showed that S-comp was associated with infiltration of T cells and monocytes as well as fibroblasts and endothelial cells, which was consistent with the pathway enrichment analysis showing a link between S-comp and angiogenesis. Conversely, E-comp was preferentially associated with natural killer (NK) cells (Fig. 5a, Supplementary Data 8A).

We also investigated the expression of immune checkpoints using our transcriptomic dataset (Fig. 5b, Supplementary Data 8A). Most immune checkpoints were positively associated with the *S*-score, including *TNFSF4* and its receptor *TNFRSF4*, *CD80*, and *PDCD1LG2* (which is more commonly referred as *PDL2*), and, more importantly, targetable immune checkpoints, such as *CD274* (*PDL1*) and *CTLA4* (Fig. 5b, c, Supplementary Figure 16). As expected, expression of these immune checkpoints showed a high correlation with the T cell infiltration score (Fig. 5b, Supplementary Data 8B). In particular, we validated the gradual change in PDL-1 protein expression according to the *S*-score level by analyzing PDL-1 expression in 30 tumor samples by reverse phase protein array (RPPA) and confirmed this association by immunohistochemical staining in a set of representative MME and MMS samples (Fig. 5d, e). Similarly, we observed a positive association between the *S*-score and expression of the immune modulator *IDO1* (Fig. 5b, c); IDO1 inhibitors are emerging in the literature and have been described as new players in the checkpoint blockade process[29]. In contrast, the E-score was positively associated mainly with two immune checkpoints *TNFSF14*, also known as *LIGHT*, and *VISTA* with high therapeutic potential[30] (Fig. 5b, c).

Taken together, these results may guide personalized immune checkpoint combination therapy in MPM according to the molecular gradients.

## Discussion

We report here one of the first multi-omics studies of MPM integrating transcriptomic and epigenetic data. We propose a new way to describe MPM heterogeneity using a bioinformatics method called WISP, which decomposes MPM molecular profiles resulting from combinations of distinct cell populations, mainly epithelioid and sarcomatoid, in addition to those that are not tumoral. This approach allows both intra-tumor and inter-tumor heterogeneity to be taken into account and is inspired by the occurrence of epithelioid and sarcomatoid morphologies in different proportions within MPM.

The WISP method relies on the assumption that distinct morphological phenotypes correspond to distinct transcriptome phenotypes. We believe that the deconvolution approach may have impacts on the potential improvement of clinical management with respect to prognosis or therapeutic strategy. The results emphasize several key points: the combination of different tumor cell components, their relationships with their micro-environments, their association with patient survival and their contribution to define new therapeutic strategies based on targeted therapies and immunotherapies.

In a recent publication, Hmeljak et al. proposed a MPM subtyping based on multi-omic data using the TCGA cohort[8]. Interestingly, their classifications based on RNA profiles (messenger/micro/long non-coding) best match their integrative classifications. This result indicates that the transcriptome is a good surrogate for MPM heterogeneity assessment. Existing molecular subtyping systems based on transcriptomic data (i.e., stratifying MPM into 2–4 subtypes) and histological types are both remarkably consistent with the WISP-derived epithelioid-like and sarcomatoid-like proportions. Using thresholds to discretize these proportions, we can quite fairly recapitulate all previous tumor stratification systems. This finding indicates that

our approach offers a more generic and finer-grained solution for describing tumor heterogeneity. In the current clinical framework, the continuum provided by our approach can be seen as problematic, unlike discrete classification systems based either on morphology or molecular parameters (e.g., shaping stratified clinical trials). However, this inconvenience can easily be solved by using thresholds. On the other hand, the future of precision medicine may benefit from the finely tuned information provided here (e.g., guidance for drug combinations and dosages to target different tumor cell compartments).

The key strength of our approach compared to existing stratifications is that it provides a very powerful method for precise analyses of tumor biology due to its ability to capture continuums. In usual subtyping systems, unrecognized mixed samples that are located between different classes will greatly impact the output of statistical tests for differential signal detection, thereby impeding underlying oncogenic pathway identification. This major limitation is absent from our approach, because WISP provides continuous proportions of the different population-based components of a tumor; these proportions can be directly correlated to various omics signals, yielding rich information for pathway analyses. These advantages are conserved when analyzing prognosis and drug response in preclinical tests.

First, we measured correlations of all genes with the E-score and S-score. As expected, the E-score and S-score were highly correlated with known biomarkers of MME and MMS, respectively. New genes were also identified, including *AXL*, which was correlated with the S-score and was of particular interest, since it was targetable by several inhibitors[13]. In addition, AXL inhibition was shown recently to enhance sensitivity to the current chemotherapeutic regimen (cisplatin and pemetrexed) in two MPM cell lines[31].

The pathway analysis revealed some additional information. In the pleura, mesothelial cells form a single layer of polarized squamous epithelial cells resting on a basement membrane. This layer is maintained by tight junctions, adherens junctions, gap junctions, and desmosomes[32]. The E-score were highly correlated with genes related to (i) cell junctions, such as claudins (*CLDN12*, *CLDN15*, and *CLDN22*), which are involved in tight junctions that link cells at their apical poles, (ii) cell polarity regulation (*PRKCI*, *PARD6B*, *F11R*, and *CRB3*) and (iii) interactions between cells and basement membranes (*LAMB3*, *ITGB4*, *KRT5*, and *CD151*). These results suggest preservation of apical/basal cell polarity in MPM with a high E-score and thus reduced aggressiveness, which is consistent with the better survival of MME compared to MMS patients. Conversely, the S-score were correlated with gene sets that generally were associated with tumor aggressiveness, including the EMT, cell cycle, hypoxia, angiogenesis, and TP53 signaling; this finding was consistent with the association of *TP53* mutations with S-comp. The S-score did not correlate with genes related to basic epithelial features but instead correlated with genes whose products either composed or were involved in communication with the basement membrane and the extracellular matrix (ECM), such as collagen-binding proteins and collagen types I, IV, and VI. These results are fully consistent with the known higher aggressiveness of MPM with a high S-score through the loss of epithelial basic features and modulation of ECM stiffness[33].

These findings shed new light on the oncogenic mechanisms at work in MPM.

After determining the E-comp-related and S-comp-related pathways based on the gene expression data, we searched for potential epigenetic regulation using both miRNome and methylome data. Very few methylome studies have focused on tumor heterogeneity, and none have integrated both methylome

and miRNome data. Our analysis identified several epigenetic biomarkers of E-comp and S-comp. The methylome data suggest strong regulation of several component-specific pathways, such as the EMT, cell adhesion, and junction pathways. Furthermore, analyses of known oncogenes and tumor suppressors revealed different methylation profiles depending on the E-score and S-score, which suggested distinct oncogenic mechanisms underlying each molecular component. miRNA-target networks highlighted candidate master regulators targeting several of the component-specific pathways. Overall, we identified potential epigenetic regulation through DNA methylation or miRNAs that might contribute to the establishment of E-comp-related and S-comp-related cell entities.

We showed that a higher S-score was related to a poorer prognosis (Hazard ratio of 6.28, P value < 0.001), even when the analysis was restricted to MME. This result showed stronger prognostic information for the S-score than histology. This result was further demonstrated by multivariate analyses showing that the S-score largely dominated the ability of the histologically defined types and the pre-existing subtyping systems[3,6] to predict survival. To facilitate clinical use of the S-score, we determined a robust cut-off for the S-score (22%) that distinguished patients with a difference in median OS of more than 10 months on average (Fig. 3b).

Analysis of the GDSC database allowed us to identify potent anticancer compounds whose efficacy was correlated with a high E-score or S-score in MPM. This correlation was validated for two of these compounds (a ROCK and a Wee1 inhibitor). GSK269962A is a selective inhibitor of the two Rho-associated protein kinases ROCK1 and ROCK2, which are the effectors of the small GTPase RhoA. The effect of GSK269962A on MPM cell viability was previously described in five commercial MPM cell lines with IC50 values ranging from 3.4 to 13.2 μM[34]. Our study identified MPM in culture with much higher sensitivity (below 1 μM) for MPM with a high S-score. ROCK inhibitors modify actin cytoskeleton dynamics, which affects diverse biological processes, including cell migration and motility, cell cycle control, cell apoptosis and cell junction integrity. However, their inhibitory effects are very dependent on the tumor cell type[35]. The higher efficiency of GSK269962A in MPM with a high S-score could be related to the probable loss of cell junctions in S-comp cells, as mentioned above. Wee1 is a crucial component of the G2–M cell cycle checkpoint that prevents entry into mitosis in response to cellular DNA damage. Impaired cell viability due to Wee1 inhibition has been shown for several cancer cell lines but not MPM. This component is most commonly used in combination with a DNA-damaging agent. Enhanced antitumor efficacy was observed in p53-deficient cell lines, including sarcomas[36,37], which could explain why MPM with a high S-score was more sensitive to Wee1 inhibition. Our data support the speculation that these two types of compounds (especially Wee1 inhibitors, which are currently in use in cancer clinical trials) may be considered for MPM with a high S-score. However, further investigations are clearly required before considering clinical applications.

The immune microenvironment of MPM is known to be heterogeneous[38]. Accordingly, the S-score and E-score are correlated with distinct immune population infiltration scores. Markers of the adaptive immune response are predominant in tumors with a high S-score, whereas markers of the innate immune response are found in tumors with a high E-score.

The correlation of the E-score with the complement pathway and NK infiltration, which are related to innate immunity, highlighted patients who were potentially suitable for antibody-based therapy by favoring complement-dependent cytotoxicity

| Table 1 Clinico-pathological characteristics and epidemiologic data of the series of MPM patients | | |
|---|---|---|
| | Exploration series (n = 63) | Validation series (n = 110) |
| Gender (n [%]) | | |
| Male | 47 [75] | 85 [77] |
| Female | 16 [25] | 25 [23] |
| Age (years) | | |
| Median ± SD | 64 ± 10 | 71 ± 11 |
| Range | 39–90 | 20–89 |
| Histology (n [%]) | | |
| Epithelioid | 49 [78] | 81 [76] |
| Biphasic | 7 [11] | 15 [8] |
| Sarcomatoid | 5 [8] | 8 [7] |
| Desmoplastic | 1 [2] | 2 [2] |
| Lymphohistiocytoid | 1 [2] | 0 [0] |
| Asbestos exposure (n [%]) | | |
| Exposed | 47 [78] | 75 [74] |
| Non-exposed | 13 [22] | 26 [26] |
| Stage IMIG (n [%]) | | |
| I | 2 [4] | 3 [6] |
| II | 7 [14] | 8 [16] |
| III | 25 [49] | 23 [45] |
| IV | 17 [33] | 17 [33] |
| Survival (months) | | |
| Median | 16.9 | 19.7 |
| Range | 0.1–165.7 | 0.4–108.5 |

*MPM* malignant pleural mesothelioma

future clinical transfer, we provide here a validated qRT-PCR gene signature that allows determination of the E-score and S-score in MPM samples.

## Methods

**Patients.** Tumor collection of frozen tumor samples included 63 cases for the exploration series and 110 cases for the qRT-PCR validation series. Frozen MPM tumor samples were retrieved from the French Mesobank collection (biobanks of two French hospitals: CHRU de Lille and CHRU de Nice) and the biobank collection of HEGP (Hôpital Européen Georges Pompidou) in Paris. Patients were diagnosed from 2003 to 2016 and certified by Mesopath as MPM. The four normal pleura were obtained after stripping and dissecting the parietal pleura from patients with lung cancer or pulmonary emphysema. The percentage of tumor cells in MPM samples was estimated by histologic examination. All patients gave their written informed consent for the use of their tumor specimen for research. This study is a part of a research project approved by a Medical Ethics Committee (CPP Ile-de-France II). All the collected samples and the associated clinical information were registered in a database (DC-2016-2771) validated by the French research ministry. Clinico-pathological and epidemiologic data of patients are reported in Table 1.

**DNA and RNA extraction.** For CIT exploration series, nucleic acids were extracted for 63 MPM tumor samples and 4 samples from normal pleura. Genomic DNA was extracted using a standard isopropanol precipitation procedure. Total RNA was extracted using trizol and Guanidine Isothiocyanate, and quality was assessed using a NanoDrop spectrophotometer (Thermo Fisher Scientific) and electrophoregram profiles on an Agilent Bioanalyzer (Agilent Technologies)[6]. For CIT validation series, total RNA was extracted for 110 MPM tumor samples using all Prep DNA RNA miRNA Universal Kit (Qiagen).

**qRT-PCR and RPPA analysis.** For mRNA expression, qRT-PCR analysis was performed using predefined TaqMan probes (Supplementary Data 1) chosen from the Thermo Fisher Scientific database (http://www.thermofisher.com). Using the High Capacity cDNA Reverse Transcription kit (Thermofisher), 1.5 μg of total RNA was reverse transcribed in a final volume of 50 μl. qPCR reactions were done using the high throughput BioMark HD system (Fluidigm) following manufacturer's instructions. Pre-amplifications of 6 ng cDNA were performed using PreAmp Master Mix (Fluidigm) with a primer mix combining each primer used in the present study except the *18S* probe due to it very high gene expression level. Expression data (Ct values) were acquired using the Fluidigm Real Time PCR Analysis software. The mean of 5 housekeeping genes (*18S*, *ACTB*, *CLTC*, *GAPDH*, and *TBP*) was used for the normalization of expression data. For protein expression, preparation of cells lysates from 30 frozen tumor samples and reverse phase protein array (RPPA) were performed as previously described[42]. Array was revealed with the anti-PD-L1 monoclonal antibody (clone E1L3N; Cell Signaling Technology).

**mRNA profiling and analysis.** mRNA profiles were obtained using GeneChip® Human Gene 2.0 ST arrays (Affymetrix), interrogating over 40,000 RefSeq transcripts, for the 67 samples of the study (63 MPM tumors and 4 samples from normal pleura). GenomEast Platform (Strasbourg, France) carried out experiments. We used the RMA algorithm (Bioconductor affy package) to normalize the data. For each gene symbol, probe with the highest expression variance was kept. The estimation of the abundance of immune cell populations infiltrating MPM was done by using MCP-counter software[28] on the gene expression dataset. mRNA dataset data is available through ArrayExpress (http://www.ebi.ac.uk/arrayexpress) under accession E-MTAB-6877.

**DNA methylation profiling and analysis.** Whole-genome DNA methylation was analyzed for 62 MPM and 6 samples of normal pleura using the Illumina Infinium HumanMethylation450 Beadchip. Integragen SA (Evry, France) carried out experiments following the manufacturer's instructions. Illumina GenomeStudio software was used to extract the beta-value DNA methylation score for each locus. We removed data from probes that contained SNPs or that overlapped with either a repetitive element or regions of insertion and deletion in the human genome. InfiniumPurify R package[43,44] was used to obtain a matrix of purified beta-values.

The CpG Island Methylator Phenotype (CIMP) index was determined using methylation Illumina Infinium HumanMethylation450 Beadchip based on previous work[45]. In brief, all CpGs in CpG islands found to be unmethylated (<30% Beta-value) in the 6 pleural samples from our series were selected. The CIMP index was calculated independently for each sample as the proportion of methylated (>30% Beta-value) CpGs among the selected normally unmethylated islands CpGs.

Methylation data are available through ArrayExpress (http://www.ebi.ac.uk/arrayexpress) under accession E-MTAB-6884.

(CDC) and antibody-dependent cell-mediated cytotoxicity (ADCC). These types of therapeutic strategies with anti-MSLN antibodies have been tested in MPM[39] and may be improved by selecting patients based on their E-score. Furthermore, inhibiting the complement system has been proposed to promote ADCC[40] and might be of interest for MPM patients with a high E-score. Remarkably, the immune checkpoint inhibitor VISTA was correlated with the E-score and was also described as being associated to the epithelioid icluster 1 in the recent TCGA publication[8]. VISTA inhibitors showed promising results in preclinical models[30] and may be of great interest for MPM patients with a high E-score. The S-score were correlated with the T cell infiltration score, which was consistent with data reported elsewhere for the Bueno sarcomatoid molecular subtype[3]. In addition, the T cell infiltration score was very highly correlated with the expression of most checkpoint inhibitors, including targetable *PDL1* and *CTLA4*. These findings suggest that MPM with a high S-score are mostly infiltrated by exhausted T cells and therefore may respond to anti-immune checkpoint therapies targeting either PDL1 or CTLA4. A recent randomized phase 2 trial (IFCT-1501 MAPS2, http://clinicaltrials.gov/show/NCT02716272) reported increased progression-free survival in MPM patients treated with anti-PDL1 without relevant response markers[41]. Determining the S-score in this series to assess its potential as a marker of response to anti-PDL1 therapy would be interesting.

The evolving technologies will permit researchers to better characterize the dynamics of the genomic evolution of tumor cell populations. Characterizing diverse cell populations using s, pecific biomarkers, microdissection or single cell analysis will be particularly interesting. Overall, this study largely renews the way of analyzing MPM biology in relation to clinical parameters, including prognosis and therapy. Importantly, our findings may guide personalized therapeutic strategies in MPM, particularly targeted therapies and immunotherapies, and highlight new avenues of investigation toward new MPM treatments. To help

**microRNA profiling and analysis**. miRNA-Seq libraries were performed using 60 MPM tumor samples with at least 1 μg of extracted total RNA with a RIN (RNA integrity number) greater than 7. Before starting, total RNAs were purified with miRNeasy kit, which allows the selection of the small RNA fraction less than 100b. From these samples enriched in small RNAs, libraries were performed according to previously established protocols[46]. The Illumina HiSeq 2000 sequencing platform generated the sequencing images. The data were analyzed in three steps: image analysis, base calling and bcl conversion. CASAVA demultiplexed multiplexed samples during the bcl conversion into compressed FASTQ files. For quality controls on raw sequence data, fastqc software was used. Finally, the script "Trim_adapter", provides by mirExpress software, handled the sequence files, which contain adapter or not according to the input of adapter sequence. The sequence adapter was trimmed on sequence data. sRNAbench[47] was used to quantify read counts for each human miRNA referenced in mirBase 20. Only mature miRNA with at least five read counts in at least 2 samples were kept for further analysis. Gene counts were normalized using the upper-quartile approach[48]. miRNA sequencing data is available through ArrayExpress (http://www.ebi.ac.uk/arrayexpress) under accession E-MTAB-6895.

To analyze miRNA gene regulations we used two databases, miRTarbase 2016[49] and TarBase V.7[50], which describe experimentally validated miRNA-target interactions. miRNA-target networks were constructed using igraph R package.

**Public datasets**. Four published datasets were used in addition to the TCGA dataset:

Reynies—Gene expression datasets were downloaded from the ArrayExpress repository under accession code E-MTAB-1719[6].

Bueno—RNAseq gene expression datasets were downloaded from the European Genome-phenome Archive under accession code EGAS00001001563. The subtype labels were available for 209 samples profiled by RNAseq[3].

Gordon—Gene expression datasets were downloaded from the provided Gene Expression Omnibus entry GSE2549[10]. All non-cancer samples were removed for tumor subtype analysis and before any gene-wise data centering.

Lopez—Gene expression datasets were downloaded from supplementary files in the related publication[9].

TCGA—TCGA data were downloaded through the Broad Institute TCGA GDAC firehose tool[11]. RNA-seq data were available for 86 samples. Unsupervised clustering revealed three outliers that were removed for further analyses.

**Genetic alterations in MPM TCGA series**. Genetic alteration data (copy number alteration and mutation) for *CDKN2A*, *NF2*, *BAP1*, and *TP53* were retrieved from cBioPortal, an online portal for accessing data from TCGA project (http://www.cbioportal.org)[51,52].

**Comparison of the classifications**. Centroid profiles were built for each classification using an approach described in previous works[53,54]. Briefly, after gene-wise centering, a centroid was built for each subtype using up to 500 most discriminant genes (ANOVA fdr adjusted $P$ values < 0.05, AUC > 0.8). The centroids for each classification are reported in Supplementary Data 9.

Centroids comparison between two datasets was performed by calculating the correlation between these centroids, restricted to the subset of genes available in both datasets.

Classifiers were built for the CIT, Bueno, and TCGA classifications based on the calculated centroids. Gene expressions of tumor samples were then correlated (Pearson's correlation) to all their centroids of a classification system (using only the subset of genes available in both datasets) and the closest centroid class with the highest correlation coefficient was assigned.

Gene expression comparison was performed by scaling all datasets by sample to allow comparison of immune populations and immune checkpoint expressions.

**Deconvolution approach**. WISP (Weighted In Silico Pathology) is a deconvolution method aiming at assessing the intra-tumor heterogeneity from bulk molecular profiles (see Code availability). In brief, WISP is a 2 steps approach that (1) automatically filters for pure entities based on predefined pure population profiles, (2) estimates the proportions of each pure population in a mixed sample. We considered a MPM sample as a mixture in various proportion of epithelioid-like, sarcomatoid-like and non-tumor components. The first step of WISP was performed on a selection of samples from our transcriptomic data (Affymetrix array on 63 tumor samples and 4 normal samples) representative of each component to generate pure population profiles defined on their specific markers (deconvolution signature). Preliminary selection of pure population profiles consisted in our series of MME samples present in subtype C1A (most extreme epithelioid subtype) for the epithelioid component, all MMS for the sarcomatoid component and normal samples available for the non-tumor component. WISP performs an iterative procedure, where it estimates for each presupposed pure sample the proportion of the different contingents and removed the samples that do not mainly contain their corresponding class (WISP default parameters). WISP method uses all the available genes to find the best markers for each pure population. A first gene filtering is based on the $P$ values of the ANOVA test comparing gene expression between all pure populations (FDR adjusted $P$ value < 0.05) and the area under the curve

(AUC) score calculated for each pure population (AUC > 0.8). For each pure population, genes are then ranked according to their expression fold change, and the top best markers is retrieved (we chose a maximum number of 50 genes per class). The resulting deconvolution signature is reported in Supplementary Data 10. WISP second step consists in weight estimation by using a non-negative least squares regression model based on the pure population profiles (centroids calculated on the pure population markers) optimized through a quadratic programming algorithm. This step was applied to the different series (CIT, Bueno, TCGA, Gordon, and Lopez) (Supplementary Data 11).

As the deconvolution signature was constructed on transcriptomic data generated from Affymetrix HG-U133 Plus 2.0 array technology, series using other technologies were scaled by sample as described in the WISP R package. WISP gives a warning when weight estimations for a sample are unreliable according to the adjusted $R^2$ and the $P$ value of the $F$-test (WISP default parameters were used). Consequently, 3 samples were removed from WISP final results in the Gordon series and 2 in the Lopez series.

We have built another MPM deconvolution signature dedicated to cell lines. Indeed, the main difference between cell lines and tissues is the presence of a complex microenvironment in tissues. Therefore, the non-tumor component based on tissue expression profiles is not well adapted to cell lines. In addition, the best markers for E-comp and S-comp defined in tissue samples are not necessarily the best in cell lines. Therefore, we used specifically for this signature the transcriptomic data of the cell lines from our previous publication[6] and applied the same approach as for tissue samples. Preliminary selection of pure population profiles consisted in MME cell lines from C1 subtype, all available MMS cell lines and 3 normal cell lines. The resulting deconvolution signature is shown in Supplementary Data 10. This deconvolution signature was used to estimate E-score and S-score in cell lines in the GDSC expression dataset[25].

We generated qRT-PCR data on a prior selection of 68 genes being highly correlated to the E-score and S-score estimated from the transcriptome in tissues and/or cell lines or known in the literature for being associated with MPM. Based on this qRT-PCR dataset, WISP method defined a 55 genes signature (Supplementary Data 1).

**Correlation and enrichment analysis**. Pearson's correlation coefficients were calculated between the E-score or S-score and the different features (gene expression, miRNA expression, CpG DNA methylation level). Correlation significance was assessed by the Pearson's correlation test as computed in the cor.test R function. $P$ values were corrected for multiple testing using FDR correction.

In order to retrieve the pathways associated to the E-score and S-score, we performed functional enrichment analyses on the genes significantly correlated to the E-score or S-score using Enrichr tool (Fisher's Exact test)[55]. We used KEGG, GO and Reactome databases. We displayed significant pathways ($P$ values < 0.05) that were retrieved by at least two of these functional databases.

**Unsupervised clustering**. Unsupervised clustering analysis was carried out on gene expression by using an extension of the ConsensusClusterPlus algorithm[56]. In brief, using all paired combinations of Pearson distance metric and linkage (Ward, complete, and average), hierarchical clustering was bootstrapped in 1000 iterations of features and samples subsampling, keeping 80% of both at each iteration. An additional level of iteration adjusted the threshold of feature variability ranging from 1% to 50% most variable probes. The consensus was given by a final hierarchical clustering using the complete linkage and the number of co-classification as sample distance.

**Survival analysis**. Overall survival was defined as the time from diagnostic to death resulting from any cause. Survival curves were estimated using the Kaplan–Meier method and compared with the log-rank test. The Cox proportional hazard regression model was used for both univariate and multivariate analyses and for estimating the hazard ratio with 95% confidence interval. Multivariate Cox analysis integrating the CIT, Gordon, TCGA, and Bueno series[3,6,10,11] was adjusted for series. Univariate and multivariate Cox regression analyses as well as Kaplan–Meier curves were computed using the survival package of the R statistical suite. Forest plot figures were drawn using the survminer R package.

In order to define a robust S-score threshold that best discriminated patient survival, we tested thresholds ranging from 0.1 to 0.5 with a step of 0.01 for 80% of samples randomly chosen in all available samples from the different series. For each threshold, the log-rank test $P$ value was calculated adjusted for series. The procedure was repeated 500 times. We generated a graph giving on the 500 simulations the average -log10($P$ value) for each tested threshold (Supplementary Figure 11). The value corresponding to the lowest $P$ value is 0.22, which appears to be the best discriminant threshold.

**Drug sensitivity analysis**. MPM in culture (17 cases), were primary cell lines established in Inserm UMRS-1138 laboratory using fresh tumor samples obtained from several French hospitals (HEGP Paris, CHRU de Lille.). Cell line authentication is based on specific gene mutations and mycoplasma contamination was tested. They were used in several previous studies showing their relevance to MPM primary tumors[6,7]. Cells were grown in RPMI 1640 culture medium, supplemented

with 10% fetal bovine serum (Gibco, Thermo Fisher Scientific), and used at low-passage numbers (<12 passages)[57]. MPM cells were seeded in triplicate on 96-well plates (Corning, Falcon) at $0.5–1 \times 10^4$ cells per well and treated with gradient concentrations of compounds diluted in DMSO (Dimethyl sulfoxide) using HP D300 Digital Dispenser (Tecan). Three compounds were used KIN001-102/Akt Inhibitor VIII (124018, Merck), 681640/Wee1 Inhibitor (681640, Merck) and a ROCK1 inhibitor, GSK269962A (S7687, Selleckchem). Cells were fixed with formaldehyde solution (252549, Sigma—4%) 72 h after treatment, stained with Hoechst 33342 (H3570, ThermoFisher—5 µg ml$^{-1}$) and imaged with a high-content imaging device (Operetta CLS, Perkin Elmer) using a 10× objective with four fields per well captured. Number of nuclei was determined using Harmony software (version 4.6, Perkin Elmer). GraphPad Prism version 6 software was used to calculate normalized area under curve (AUC) and IC50 (GI50) of inhibitors assays.

**Immunohistochemical staining**. Immunohistochemical protocol was adapted from Ilie et al.[58]. Specimens were sectioned at a thickness of 3 µm and stained on positively charged glass slides. Deparaffinization, rehydration, and antigen retrieval were performed by CC1 (prediluted; pH 8.0) antigen retrieval solution (Ventana Medical Systems, Inc.), performed on the VENTANA BenchMark ULTRA automated slide stainer for 32 min at 100 °C. Specimens were incubated with primary antibodies as noted in Supplementary Table 1 followed by visualization with the OptiView DAB IHC Detection Kit (Ventana) and OptiView Amplification Kit (Ventana) for 12 min for PD-L1 detection. The specimens were then counterstained with haematoxylin II and bluing reagent (Ventana) and coverslipped. Each IHC run contained a positive control (on-slide placenta tissue for PD-L1). Morphological characteristics and size of the nucleus were taken into account to estimate the labeling of fibroblasts by vimentin antibodies.

## Code availability
WISP is freely available at: https://cit-bioinfo.github.io/WISP/.

## Data availability
The mRNA expression data, DNA methylation data and miRNA sequencing data have been deposited in ArrayExpress database under accession codes E-MTAB-6877, E-MTAB-6884 and E-MTAB-6895, respectively.

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

## Acknowledgements

This work is part of the national program Cartes d'Identité des Tumeurs (CIT) funded and developed by the Ligue Nationale Contre le Cancer. This work was also supported by Inserm, the Ile-de-France committee of the Ligue Contre le Cancer, the Chancellerie des Universités de Paris (Legs POIX). FM was supported by grants from the Institut thématique multi-organismes (ITMO) Cancer (Plan Cancer 2014–2019) and the Société Française de Chirurgie—Thoracique et Cardiovasculaire (SFCTV). LQ was supported by grants from Cancéropôle Région Île-de-France. The Inserm research unit is supported by the Labex OncoImmunology (investissement d'avenir), Coup d'Elan de la Fondation Bettencourt-Shueller, the SIRIC CARPEM, the Institut thématique multi-organismes (ITMO) Cancer (Plan Cancer 2014–2019) and Cancéropôle Île-de-France. The Institut Curie RPPA platform is supported by Cancéropôle Ile-de-France and we thank Bérengère Ouine et Aurélie Cartier for acquisition of RPPA data. English was improved by Nature Research Editing Service from Springer Nature (http://authorservices.springernature.com/language-editing/).

## Author contributions

L.A., M.A., C.M., L.Q., D.T., A.R., and R.T. performed sample preparation and the biological experiments. M.C.C. performed the histochemical staining and immunohistochemical scoring. L.dK. was responsible for the RPPA experiments. M.A. and N.E. conducted data management. F.M., M.C.C., P.H., V.H., H.P., and F.L.P.B. provided material and clinical annotations. Y.B. performed bioinformatics analyses. Y.B., D.J., M.C.J., A.dR., C.M., and J.Z.R. contributed to the experimental design, data analysis and discussion. D.J., M.C.J., and A.dR. directed the project. Y.B., D.J., M.C.J., and A.dR. wrote the manuscript.

## Additional information

**Competing interests:** The authors declare no competing interests.

