## [Peer Review File · Nature Communications]

Reviewers' comments:

Reviewer #1 (Remarks to the Author):

Comments on:

Dissecting heterogeneity in malignant pleural mesothelioma through histo-molecular gradients for clinical applications

1. Please provide the genes used to calculate the signature, and the precise methodology for this. Are all genes considered equal or is there weighting? Are raw values used, or z scores? etc. The methods state: Based on a prior selection of 68 genes, WISP method defined a 55 genes signature based on qRT-PCR dataset. This is not adequate in explaining how these genes were identified and incorporated in the WISP model.

Suggest that you could consider a change in terminology from E-comp and S-comp, to simple S score, which ranges from 0 to 100, reflecting the S component.

2. "Interestingly, 110 genes identified ..." I would say "As expected, ..."

3. "Our study also highlighted new genes such as PDZK1IP1 and AXL that were linked to E-comp or S-comp..." Indicate which (E vs. S) is the association for each gene for higher expression.

4. "Then, we characterized E-comp and S-comp at the epigenetic level using methylome and miRNome data." Every time you make a reference to a data set, you must indicate the origin, whether new to this study or from another source. If the latter, then give citation. Also give details. How many samples analyzed by what method in this specific case?

5. "compared to in the"

6. "This result suggests that E-comp and S-comp are related to distinct oncogenic mechanisms" But the main thesis of the paper is that there is a continuous spectrum from E to S in mesothelioma; then there should not be distinct oncogenic mechanisms.

7. "Then, we characterized E-comp and S-comp at the epigenetic level using methylome and miRNome data." miRNA expression is not generally considered an epigenetic phenomenon.
8. "we integrated our miRNA dataset ..." As above, define each data set in terms of provenance, sample size, method of assessment when you discuss it.
9. For the section, MPM molecular gradients are associated with prognosis and drug sensitivity, please show survival curves for the pooled meso data according to levels of the S-comp score, rather than just binary <20% vs. >20%. E.g. in deciles of values; or 0-10%, 10-20%, etc. Provide evidence that 20% is the best discriminant.
10. Please provide evidence that the MCP-counter tool works robustly on meso data sets, since it is quite possible that sarcomatoid meso has expression characteristics that are similar to the canonical fibroblast marker gene set used in this analysis, leading to confusion in assessment of fibroblast infiltration. Ideally, this tool should be applied to a series of mesothelioma cases for which both bulk RNA-seq and single cell RNA-seq data are available as a test case.
11. Figure 5b. What do the expression values mean? Please give both p and fdr-corrected q values for all correlations, rather than the simple green for $p < 0.05$.
12. Figure 5c, do not fade out the low S-comp score samples at left - makes it hard to see them. The x axis values make it clear they have low values.
13. "We report here the first multi-omics study of MPM integrating transcriptomic and epigenetic data." The TCGA publication is coming out soon, so this will not be true for long, would acknowledge this, or not try to claim priority.
14. "The deconvolution approach allows discussion of evolving concepts in MPM development" ??
15. "Existing molecular subtyping systems (i.e., stratifying MPM into 2 to 4 subtypes) and histological types are both remarkably consistent with the WISP-derived epithelioid-like and sarcomatoid-like proportions." So then is the WISP classification scheme really an advance, or just a slightly different way of looking at things?

16. "These findings shed new light on the oncogenic mechanisms at work in MPM." As above, if mesothelioma is a continuum, then do different pathogenic mechanisms really apply? Or is this continuum just a matter of late stage tumor evolution, which does not reflect oncogenesis?

17. "To facilitate clinical use of the S-comp score, we determined a robust cut-off for S-comp (20%) that distinguished patients with a difference in median OS of more than 10 months on average." As above, not shown as far as I can tell.

18. "VISTA inhibitors are emerging " Sadly human clinical trials of the single clinical-grade antibody against VISTA have been stopped for reasons known only to the pharma company. So this is probably not true, or is at least an overstatement.

19. Figure S10c. It is surprising that Bueno sarcomatoid subtype has a worse prognosis than any other subtype. Was there an error in generation of this figure?

Reviewer #2 (Remarks to the Author):

The authors used a new method called WISP to estimate the proportions of the epithelioid-like and sarcomatoid-like molecular components (E-comp and S-comp, respectively) in Malignant pleural mesothelioma (MPM). These proportions are highly correlated with histology among other features.

They identified gene expression, methylation and miRNA that correlate with E-comp and S-comp proportions

They found that E-comp and S-comp predict survival. The sensitivity to 22 compounds also correlated with E-comp and S-comp

Finally E-comp and S-comp correlated with inferred immune population abundance from transcriptomics, for example S-comp was associated with predicted infiltration of T cells and monocytes as well as fibroblasts and endothelial cells.

Overall this is an observational but sophisticated, integrative analysis of MPM profiles. A natural orthogonal analysis would be a single cell omic one but I understand that this is out of scope. Validation of immune and stromal infiltration predictions using IHC would be a plus. One key missing aspect is an analysis of mutations and how they correlate with E-comp and S-comp proportions. I am also unclear as to how the different components are maintained in cell lines, which would be

expected to be less heterogeneous than tissues. Commenting on these aspects in the paper would be important.

Reviewer #3 (Remarks to the Author):

The manuscript “Dissecting heterogeneity in MPM through histo-molecular gradients for clinical applications” by Blum and colleagues claims to shed new light on the intra-tumor heterogeneity which could lead to a reconsideration of MPM molecular classification which may have implications for treatment.

Molecular classification of common tumour-types like breast and lung cancer has revealed subtypes that are more susceptible to specific treatments. MPM is a relatively rare tumor-type with currently limited effective treatment options. The current study adds to the previous gene expression based data sets publically available (and cited in the paper) and the clinico-pathology literature to confirm that the underlying histology of MPM predicts clinical behaviour. Showing that the gene expression profile of MPM tumours reflect the gradient in histologies seen by pathologists. The structured clinical application of this information has always been difficult in part because of limited case numbers and treatment options. There are a number of validated prognostic models based upon physical parameters (performance status, weight loss etc) and simple biochemical and haematological factors that are useful for prognostication and selecting treatment options; it is unlikely that, at present, clinicians could justify an additional molecular characterisation of a tumour to change treatment decisions based upon E-comp or S-comp.

The reporting of compounds that may be effective against sarcomatoid mesothelioma is a welcome finding.

This study uses the software, WISP, available on github to apply a weighting to a set of genes generating an output of two numbers (1) epithelioid component and (2) the sarcomatoid component. I note that having the software hosted by github comes with its imprimatur (i.e. good quality, stable software) – however there appears to be no associated publication or validation of the software. Could the authors provide more justification for the use of this software package for their study? From data presented in Table S10 from the WISP package, the combined value of the two output data sets for each sample has a mean of 0.9045 (95% CI 0.89-92) for the 437 cases.

Specifically 67% of the samples analysed are reported as being comprised of genes reflecting 90% tumour content. (However, one sample has a combined final total of 0.0997; ie less than 0.1% of genes reflecting tumour content): (Conversely, I note that 5 samples were removed from the analyses – presumably because their combined value was greater than 1). There is no discussion, or presentation of these issues in the main manuscript. Would the authors like to comment on the effect of the tumour purity of the initial sample and how this would affect analysis; are there assumptions made by the WISP program about multiple cell types in the sample? This is important to note in light of subsequent analysis using the MCP counter package.

- From a minor technical point-of-view is it appropriate to include the 8 epithelioid and 3 sarcomatoid samples used to train the model (step 1) in the estimation (step 2)?

- Could the methods (page 27) be clarified, as I understand 152 genes (Table S9) were in the original deconvolution signature, however it is not clear to me what the 68 genes were “based on a prior selection”. (Similarly, were the qRT-PCR gene lists determined before or after the WISP analysis? as written it appears that the genes used to validate the WISP defined gene set were already known “...WISP method defined a 55 genes signature based on qRT-PCR dataset (Table S1) – this needs to be clarified).

- Page 27 - the separate analysis on the cell line data to construct “a more appropriate” signature needs to be better justified. I assume this is to reflect the relative purity of cell line samples versus tumour tissue – which ties in with my previous comment.

- Minor point – Results Page 3 – Supp Fig 1 does not refer to prognosis but rather Supp Fig 2 – which is not referred to in the text.

- For Fig 1b – is it correct that data is present across the whole sample set of 442 samples for the subsets rows CIT; Bueno and TCGA.

From a perspective of readability, the authors may consider for a publication which has the “Results” section before the “Methods” a bit more description in the results text on how analysis was done and on what samples.

Point-by-point response to the referees' comments

First, we would like to thank all the three reviewers for their precious inputs. Please find below a point-by-point response to each of the comments.

Reviewers' comments:

Reviewer #1 (Remarks to the Author):

Comments on:

Dissecting heterogeneity in malignant pleural mesothelioma through histo-molecular gradients for clinical applications

We thank the reviewer for all his/her fruitful comments.

1a. Please provide the genes used to calculate the signature, and the precise methodology for this.

We are sorry we did not provide sufficient description of this step. WISP method selects among all available genes the best markers for each user-defined cell population; here we considered three populations: epithelioid, sarcomatoid and non-tumor. WISP needs representative samples of these populations to identify related markers. WISP includes a preliminary step that removes atypical representative samples (if any), thus remaining with pure representative samples of each population. A first gene filtering is based on the P-value of the ANOVA test comparing gene expression between all pure populations (FDR adjusted P-value<0.05) and the area under the curve (AUC) score calculated for each pure population

(AUC>0.8). For each pure population, genes are then ranked according to their expression fold change, and the top best markers is retrieved (we chose a maximum number of 50 genes per class). We have added this clarification in the method section “Deconvolution approach”. The genes that are part of the deconvolution signatures were already given in Supplementary Table 9 (now Supplementary Table 10).

“Preliminary selection of pure population profiles consisted of MME samples present in subtype C1A (most extreme epithelioid subtype), all MMS and normal samples available in our series. WISP performs an iterative procedure where it estimates for each presupposed pure sample the proportion of the different contingents and removed the samples that do not mainly contain their corresponding class (WISP default parameters). WISP method uses all the available genes to find the best markers for each pure population. A first gene filtering is based on the P values of the ANOVA test comparing gene expression between all pure populations (FDR adjusted P value<0.05) and the area under the curve (AUC) score calculated for each pure population (AUC>0.8). For each pure population, genes are then ranked according to their expression fold change, and the top best markers is retrieved (we chose a maximum number of 50 genes per class). The resulting deconvolution signature is reported in Supplementary Table 10.”

1b. Are all genes considered equal or is there weighting? Are raw values used, or z scores? etc.

There is no gene weighting strictly speaking. The centroids of the pure populations on their best markers (deconvolution signature) are calculated and injected into a non-negative least squares regression model for the weight estimations. To answer the second question, normalized expression values (expression data were normalized by the RMA normalization

procedure as explained in the method section “mRNA profiling and analysis”) are used to calculate the centroids.

We added this clarification in the method section “Deconvolution approach”.

“WISP second step consists in weight estimation by using a non-negative least squares regression model based on the pure population profiles (centroids calculated on the pure population markers) optimized through a quadratic programming algorithm. This step was applied to the different series [...]”

1c. The methods state: Based on a prior selection of 68 genes, WISP method defined a 55 genes signature based on qRT-PCR dataset. This is not adequate in explaining how these genes were identified and incorporated in the WISP model.

With the primary objective of validating the expression of genes of interest by qRT-PCR, we have made a selection of genes being highly correlated to the E-score and S-score estimated from the transcriptome in tissues and/or cell lines, and have also added genes well known for being associated with MPM such as *BAP1*, *CDKN2A* and *KRT6A*. This selection corresponds to 68 genes described in Supplementary Table 1.

In order to create a deconvolution signature in qRT-PCR, we applied WISP on this dataset. As described above (see point 1a), based on the available genes (here 68 genes), WISP selects the best markers for each component (epithelioid, sarcomatoid and non-tumor components) which resulted here in a 55 genes signature. The centroids of the pure populations were then calculated from the qRT-PCR measurements on these 55 genes and integrated into the regression model for weight estimations. As shown in Supplementary Fig. 5, the weights estimated from the qRT-PCR data using this signature are very strongly correlated to those estimated from the transcriptome microarray data (affymetrix) using the

transcriptomic-based 150 genes signature. This result supports the relevance of the 55 genes signature.

We have modified the text accordingly to provide more clarity on this prior gene selection for qRT-PCR data.

“We generated qRT-PCR data on a prior selection of 68 genes being highly correlated to the E-score and S-score estimated from the transcriptome in tissues and/or cell lines or known in the literature for being associated with MPM. Based on this qRT-PCR dataset, WISP method defined a 55 genes signature (Supplementary Table 1).”

1d. Suggest that you could consider a change in terminology from E-comp and S-comp, to simple S score, which ranges from 0 to 100, reflecting the S component.

E-comp and S-comp terminology corresponds to the name of the epithelioid-like and sarcomatoid-like components that we defined. A score will rather correspond to the proportion of any of these components. For this reason, and to take into account the reviewer's comment, we defined E-score and S-score as the E-comp and S-comp proportions, respectively.

Both definitions are now given in the main text:

“[...] the proportions of the epithelioid-like and sarcomatoid-like molecular components after preselection of pure population profiles. We named these molecular components E-comp and S-comp to distinguish them from the epithelioid and sarcomatoid histologically defined morphologies and their proportions in a given sample as the E-score and S-score, respectively.”

2. "Interestingly, 110 genes identified ..." I would say "As expected, ..."

We took into account this suggestion and the text was modified accordingly in the manuscript.

"As expected, 110 genes identified in other studies as overexpressed in MME or in MMS, including *UPK3B*, *MSLN*, *CLDN15*, *LOXL2* and *VIM*, were significantly positively correlated with the E-score or S-scores proportions^{3,9}."

3. "Our study also highlighted new genes such as PDZK1IP1 and AXL that were linked to E-comp or S-comp..." Indicate which (E vs. S) is the association for each gene for higher expression.

The associations for each gene are now clarified in the text.

"Our study also highlighted new genes such as PDZK1IP1 and AXL that were positively correlated to the E-score or S-score, respectively."

4. "Then, we characterized E-comp and S-comp at the epigenetic level using methylome and miRNome data. " Every time you make a reference to a data set, you must indicate the origin, whether new to this study or from another source. If the latter, then give citation. Also give details. How many samples analyzed by what method in this specific case?"

These precisions are now mentioned in the text.

"Next, we characterized E-comp and S-comp at the epigenetic level using methylome data and at the miRNA level using miRNome data. These methylome and miRNome data were

generated from 62 and 60 MPM tumor samples included in our Affymetrix dataset (CIT exploration series), respectively.”

We also specified the origins of dataset for immune analysis.

“In this context, we analyzed immune populations using the Microenvironment Cell Population Counter (MCP-counter) tool on our transcriptomic dataset²⁵.”

“We also investigated the expression of immune checkpoints using our transcriptomic dataset (Fig. 5b, Supplementary Table 8.A).”

5. "compared to in the"

This typographic error was corrected.

“Surprisingly, among the genes known to be frequently hypermethylated in MPM compared to the normal pleura[...] “

6. "This result suggests that E-comp and S-comp are related to distinct oncogenic mechanisms" But the main thesis of the paper is that there is a continuous spectrum from E to S in mesothelioma; then there should not be distinct oncogenic mechanisms.

Our aim is to describe the intra-tumor heterogeneity of MPM, meaning that we expect distinct oncogenic mechanisms, driving distinct tumor cell types, to coexist within some MPM samples. Our data identified oncogenes and tumor suppressor genes whose methylation and expression levels show negative or positive correlation with E-comp or S-comp proportions. This supports that E-comp or S-comp proceed from distinct oncogenic mechanisms, while both components can be found within the same tumor.

To clarify this point, we modified the sentence as follows:

"This result suggests that distinct (intra-cellular) oncogenic mechanisms are present in E-comp or S-comp related cells."

7. "Then, we characterized E-comp and S-comp at the epigenetic level using methylome and miRNome data." miRNA expression is not generally considered an epigenetic phenomenon.

Even if miRNAs are sometimes considered as part of the epigenetic mechanism, we followed the advice of the reviewer and modified the text accordingly.

"Next, we characterized E-comp and S-comp at the epigenetic level using methylome data and at the miRNA level using miRNome data."

8. "we integrated our miRNA dataset ..." As above, define each data set in terms of provenance, sample size, method of assessment when you discuss it.

Provenance and sample size are now mentioned in the text. Methods of assessment were already indicated in the Material and Method section.

"In addition, we integrated our miRNA dataset, obtained from 60 MPM included in our Affymetrix dataset (CIT exploration series) [...]"

9. For the section, MPM molecular gradients are associated with prognosis and drug sensitivity, please show survival curves for the pooled meso data according to levels of the

S-comp score, rather than just binary <20% vs. >20%. E.g. in deciles of values; or 0-10%, 10-20%, etc. Provide evidence that 20% is the best discriminant.

In order to facilitate the clinical use of the S-score, we looked for a robust S-score threshold above which, the S-comp has a particularly deleterious effect on survival.

The reviewer suggests us to look at various intervals such as [0-10] or [10-20]% of S-score instead of considering a binary comparison. To investigate the reviewer suggestion on intervals, we first generated the survival curves for all the different intervals [0-10], [10-20]...[90-100] (see below).

The figure shows the gradual association of the S-score with a poor prognosis as it is already demonstrated by the cox model that we showed in our study (see figure 3d).

As described in the material and method section, we have tested various thresholds ranging from 0.1 to 0.5 with a step of 0.01 and used a bootstrap procedure (80% of samples randomly chosen in all available samples from the different series at each of the 500 iterations) to obtain a more robust threshold. The final threshold corresponds to the average of the thresholds associated to the lowest P-value at each iteration and was equal to 0.20.

To answer the reviewer's request, we generated a graph giving on the 500 simulations the average $-\log_{10}(P\text{-value})$ for each tested threshold.

Supplementary Figure 11

Determination of the best survival discriminant threshold of the S-score.

For each tested threshold ranging from 0.1 to 0.5 with a step of 0.01, the average $-\log_{10}(P\text{-value})$ over the 500 simulations is represented.

We can see from this graph that the threshold of 0.2 corresponds actually to a local minimum, which we didn't notice by considering the average value of all the 500 best thresholds. The value corresponding to the lowest P-value is 0.22, which appears to be the best discriminant threshold.

We thank the reviewer for his/her comment that allowed us to improve this analysis.

We therefore have regenerated all the survival curves with this new threshold. All the conclusions remain unchanged. We also modified the method section and added the generated graph as a supplementary figure 11.

“In order to define a robust S-score threshold that best discriminated patient survival, we tested thresholds ranging from 0.1 to 0.5 with a step of 0.01 for 80% of samples randomly chosen in all available samples from the different series. For each threshold, the log-rank test P value was calculated adjusted for series. The procedure was repeated 500 times. We generated a graph giving on the 500 simulations the average $-\log_{10}(P \text{ value})$ for each tested threshold (Supplementary Fig. 11). The value corresponding to the lowest P value is 0.22, which appears to be the best discriminant threshold.”

10. Please provide evidence that the MCP-counter tool works robustly on meso data sets, since it is quite possible that sarcomatoid meso has expression characteristics that are similar to the canonical fibroblast marker gene set used in this analysis, leading to confusion in assessment of fibroblast infiltration. Ideally, this tool should be applied to a series of mesothelioma cases for which both bulk RNA-seq and single cell RNA-seq data are available as a test case.

We thank the reviewer for this interesting comment.

MCP-counter is a robust method for quantifying the absolute abundance of eight immune and two stromal cell populations in heterogeneous tissues from transcriptomic data. It has been validated using a validation series of over 4800 transcriptomes of sorted cell populations and tumor cell lines, *in vitro* mRNA mixtures and *ex vivo* immunohistochemical data from colorectal cancer tumors (Becht et al, Genome Biology, 2016, PMID: 27765066). Concerning the eventual confusion between sarcomatoid MPM and fibroblasts, it is important to note that among the MCP-counter fibroblast signature, only one of the 8 markers is part of our deconvolution signature as a sarcomatoid marker (*GREM1*), suggesting that these two entities can be properly distinguished. In addition, to check the possibility of confounded signatures for MPM and fibroblasts we first reanalyzed the Affymetrix HG-U133plus2 expression data (n=1936, training series) from (Becht et al,

Genome Biology, 2016, PMID: 27765066) which includes mesothelioma cell lines (n=9, including 6 epithelioids, 2 biphasic and 1 sarcomatoid) and a diversity of samples of sorted cells, including fibroblasts (n=50), endothelial cells (n=18) and T cells (n=400). These data are normalized with single sample frozen RMA procedure, which reduces batch effects. Principal Components Analysis shows that mesothelioma cell lines (M) are well separated from fibroblasts (F), endothelial (E) or T cells (T).

In response to the reviewer's comment on the accuracy of the MCP-counter estimations in MPM, there are unfortunately no public data available with bulk and single cell RNAseq profiles. Nevertheless, we illustrated as well MPCcounter performance on a public dataset consisting in sorted cells from lung tumors, where pan-immune, fibroblast and endothelial cells were isolated (retrieved from GEO:GSE111907). As shown in the heatmap of the MCP-counter scores, we can see the relevance of the estimations in particular for fibroblasts and endothelial cells. In our study, we showed a positive correlation between the sarcomatoid component and the T-cell and monocyte immune populations. Interestingly, Bueno et al.

showed similar results (Bueno et al. 2016, PMID 26928227), which reinforces our results and MCP-counter estimation accuracy in MPM.

In addition, we performed immunohistochemistry (IHC) on immune and stromal populations to test the accuracy of MCP-counter estimations in MPM. Presences of monocytic lineage cells, T cells, CD8 T-cells and fibroblasts were evaluated on 16 representative samples (high, intermediate and low MCP-counter scores) using CD163, CD3, CD8 and vimentin specific antibodies, respectively. Morphological characteristics and size of the nucleus were taken into account to estimate the labeling of fibroblasts by vimentin antibodies. As shown below, we obtained significant correlations between IHC estimations and MCP-counter scores which validates the accuracy of MCP-counter in MPM.

Immunohistochemistry (IHC) on immune and stromal populations.

a Correlation between IHC staining scores and MCP-counter scores. Red dots correspond to samples for which corresponding immunostaining picture is shown.

b Representative pictures of stained FFPE samples.

These results also address the reviewer's concerns about fibroblasts assessment.

Correlation between IHC staining and MCP-counter scores, and representative picture of stained FFPE samples are now added as a supplementary figure (Supplementary figure 15).

We added the following sentences in the Results section:

“We validated the performance of MCP-counter in MPM by immunohistochemistry (IHC) on immune and stromal populations (Supplementary fig.15). Correlation analyses between the S-score and MCP-counter estimations [...]”

The Methods section was modified accordingly:

“Specimens were incubated with primary antibodies as noted in Supplementary Table 12 followed by visualization with the OptiView DAB IHC Detection Kit (Ventana) and OptiView Amplification Kit (Ventana) for 12 minutes for PD-L1 detection. The specimens were then counterstained with haematoxylin II and bluing reagent (Ventana) and coverslipped. Each IHC run contained a positive control (on-slide placenta tissue for PD-L1). Morphological characteristics and size of the nucleus were taken into account to estimate the labeling of fibroblasts by vimentin antibodies.”

11. Figure 5b. What do the expression values mean? Please give both p and fdr-corrected q values for all correlations, rather than the simple green for $p < 0.05$.

By “expression” we meant both the immune population scores and the immune checkpoint (ICK) gene expressions. To avoid any confusion, we modified the figure in order to have two different legends of the heatmap color scale: one referring to the MCP-counter score and one to the ICK gene expression.

For a sake of clarity of the figure, we provide the correlations and associated P-values and FDR adjusted P-values as a new Supplementary Table (Supplementary Table 8) showing the correlation between immune population scores or ICK gene expressions and E-score or S-score (Supplementary Table 8.A) and the correlation between ICK gene expression and T-cell MCP-counter score (Supplementary Table 8.B) .

12. Figure 5c, do not fade out the low S-comp score samples at left - makes it hard to see them. The x axis values make it clear they have low values.

We thank the reviewer for his/her judicious comment. We agree that the light green color used for the low S-score values made the corresponding dots difficult to see in this figure.

We have removed the color gradients from Figure 5c,d. We modified accordingly the Supplementary Figure 16.

The legend of Figure 5c,d was modified : “For each plot, the correlation and P value are shown (Pearson’s correlation test). The point shapes correspond to the different MPM histologies: epithelioid (E), biphasic (B), sarcomatoid (S) and desmoplastic (Desmo).”

13. "We report here the first multi-omics study of MPM integrating transcriptomic and epigenetic data." The TCGA publication is coming out soon, so this will not be true for long, would acknowledge this, or not try to claim priority.

We modified the sentence accordingly:

“We report here one of the first multi-omics studies of MPM integrating transcriptomic and epigenetic data.”

14. "The deconvolution approach allows discussion of evolving concepts in MPM development" ??

We agree that this sentence was too speculative as our study does not provide direct evidence on tumor development. Consequently, we modified accordingly the sentence in the discussion:

"We believe that the deconvolution approach may have impacts on the potential improvement of clinical management with respect to prognosis or therapeutic strategy."

15. "Existing molecular subtyping systems (i.e., stratifying MPM into 2 to 4 subtypes) and histological types are both remarkably consistent with the WISP-derived epithelioid-like and sarcomatoid-like proportions." So then is the WISP classification scheme really an advance, or just a slightly different way of looking at things?

WISP should not be considered as a classification method strictly speaking as it provides continuous proportions of the different population-based components of a tumor.

We believe that WISP approach is a step forward to better understand MPM for several reasons. First, contrary to hierarchical classification, it does not rely on a strict subtype assignation and allows to take into account intermediate phenotypes. The continuum offers a more precise solution for describing tumor heterogeneity. Second, this approach offers a better description of intra-tumor heterogeneity. It associates molecular components to tumors heterogeneous morphological features. It also allows to better consider the heterogeneity of the tumor microenvironment. Third, at the clinical level, the continuous proportions of the different population-based components of a tumor can be directly correlated to patient survival. We showed that robust threshold can be defined leading to a better prediction of patient survival than existing tumor classification systems. Furthermore, in the future, robust threshold could be also defined for prediction of patient response to

treatment. Precision medicine may also benefit from these scores by defining drug combinations and improving dosages to target different tumor cell compartments.

16. "These findings shed new light on the oncogenic mechanisms at work in MPM." As above, if mesothelioma is a continuum, then do different pathogenic mechanisms really apply? Or is this continuum just a matter of late stage tumor evolution, which does not reflect oncogenesis?

As the transcriptomic analysis is carried out in MPM tumors from different patients, the oncogenic mechanism can be seen as the differential expressions of unbalanced genes, which move forward to progress in the oncogenic process. This model is compatible with a regulatory role of miRNAs and methylation, and consistent with the observation of the differential expression of oncogenes and genes playing a role on the organization of the tumor microenvironment. Moreover, as shown by the analyses of tumors from different patients, we cannot really refer to an early or late stage, because we do not have samples at different stages from a given patient. As far as we know, the stages of MPM progression are not as well-known as for other cancers.

17. "To facilitate clinical use of the S-comp score, we determined a robust cut-off for S-comp (20%) that distinguished patients with a difference in median OS of more than 10 months on average." As above, not shown as far as I can tell.

The difference in median OS is now added on the plots of the Figure 3b and mentioned in the legend of that Figure.

"Difference in median overall survival between patients with less or more 22% of S-score is indicated in red (b). "

We also mentioned the Figure 3b in the related statement in the discussion

“To facilitate clinical use of the S-score, we determined a robust cut-off for the S-score (22%) that distinguished patients with a difference in median OS of more than 10 months on average (Figure 3b).”

18. "VISTA inhibitors are emerging " Sadly human clinical trials of the single clinical-grade antibody against VISTA have been stopped for reasons known only to the pharma company. So this is probably not true, or is at least an overstatement.

We thank the reviewer for alerting us that VISTA inhibitor with clinical application has been stopped by the pharma company. Consequently, we just highlighted promising result of VISTA inhibitor in preclinical models.

“VISTA inhibitors showed promising results in preclinical models²⁸ and may be of great interest for MPM patients with high E-score”

19. Figure S10c. It is surprising that Bueno sarcomatoid subtype has a worse prognosis than any other subtype. Was there an error in generation of this figure?

There was no error in generating the figure, but we thank the reviewer for his/her careful observation.

Indeed, if we only consider the Bueno classification in the cox model, the sarcomatoid cluster has, as expected, the worst prognosis (see the Figure below).

Variable	N	Hazard ratio	p
pred.bueno Epithelioid	120	Reference	
Biphasic-E	97	2.19 (1.59, 3.02)	<0.001
Biphasic-S	90	2.15 (1.54, 2.99)	<0.001
Sarcomatoid	75	2.45 (1.75, 3.42)	<0.001

The goal of our analysis was to compare the prognostic value of our scores (E-score and S-score) and the one of Bueno's clusters. That is why in our multivariate model, we integrated the two systems. It seems that the S-score has the highest prognostic value and therefore captures the largest explanatory part in the model (Figure S10c, now S12c). The low HR value of the sarcomatoid cluster is due here to the small explanatory part remaining in the model and its collinearity with the S-score (both being associated to a poor prognosis).

Reviewer #2 (Remarks to the Author):

The authors used a new method called WISP to estimate the proportions of the epithelioid-like and sarcomatoid-like molecular components (E-comp and S-comp, respectively) in Malignant pleural mesothelioma (MPM). These proportions are highly correlated with histology among other features.

They identified gene expression, methylation and miRNA that correlate with E-comp and S-comp proportions

They found that E-comp and S-comp predict survival. The sensitivity to 22 compounds also correlated with E-comp and S-comp

Finally E-comp and S-comp correlated with inferred immune population abundance from transcriptomics, for example S-comp was associated with predicted infiltration of T cells and monocytes as well as fibroblasts and endothelial cells.

1. Overall this is an observational but sophisticated, integrative analysis of MPM profiles. A natural orthogonal analysis would be a single cell omic one but I understand that this is out of scope.

We completely agree with the reviewer that it would be of particular interest to generate single cell data in MPM, but this is out of the scope of our study. We already mentioned this point in our discussion: “The evolving technologies will permit researchers to better characterize the dynamics of the genomic evolution of tumor cell populations. Characterizing diverse cell populations using specific biomarkers, microdissection or single cell analysis will be particularly interesting.”

2. Validation of immune and stromal infiltration predictions using IHC would be a plus.

Of note, as a request of reviewer 1, the proportions of E-comp and S-comp in a given sample were named E-score and S-score, respectively.

As also explained in our answer to the point 10 raised by reviewer #1, MCP-counter is a robust method for quantifying the absolute abundance of eight immune and two stromal cell populations in heterogeneous tissues from transcriptomic data, which has been validated using a large transcriptome validation series (n>4800), in vitro mRNA mixture and ex vivo immunohistochemical data from colorectal cancer tumors (Becht et al, Genome Biology, 2016, PMID: 27765066).

We performed immunohistochemistry (IHC) on immune and stromal populations to test the accuracy of MCP-counter estimations in MPM. Presences of monocytic lineage cells, T cells,

CD8 T-cells and fibroblasts were evaluated on 16 representative samples (high, intermediate and low MCP-counter scores) using CD163, CD3, CD8 and vimentin specific antibodies, respectively. Morphological characteristics and size of the nucleus were taken into account to estimate the labeling of fibroblasts by vimentin antibodies. As shown below, we obtained significant correlations between IHC estimations and MCP-counter scores which validates the accuracy of MCP-counter in MPM.

Immunohistochemistry (IHC) on immune and stromal populations.

a Correlation between IHC staining scores and MCP-counter scores. Red dots correspond to samples for which immunostaining picture is shown. **b** Representative pictures of stained FFPE samples.

Correlation between IHC staining and MCP-counter scores, and representative picture of stained FFPE samples are now added as a supplementary figure (Supplementary figure 15).

We added the following sentences in the Results section:

“We validated the performance of MCP-counter in MPM by immunohistochemistry (IHC) on immune and stromal populations (Supplementary fig.15). Correlation analyses between the S-score and MCP-counter estimations [...]”

3. One key missing aspect is an analysis of mutations and how they correlate with E-comp and S-comp proportions.

Using available mutation and copy number alteration (CNA) data from TCGA retrieved using cBioportal, we performed the analysis of the associations between mutations and E-comp or S-comp. We focused on key genes altered during mesothelial carcinogenesis: *CDKN2A*, *NF2*, *BAP1* and *TP53*. As shown in the Figure below, we observed positive correlations of *NF2* and *TP53* genetic alterations with the S-score.

Supplementary Figure 7

Association between E-comp and/or S-comp and genetic alterations

Genetic alterations in well-known altered genes in MPM, including point mutations and copy number alterations (CNA) in the TCGA series. Lateral bars on the right correspond to $-\log_{10}(P \text{ value})$ of the t.test comparing for a specific gene the E-score (brown) or the S-score (green) between samples with or without any alterations. The grey dashed line corresponds

to a *P* value threshold of 0.05. Gene labels are colored in red when at least one of the corresponding tests is significant (*P* value<0.05).

We took into account these new results in the manuscript by including this figure as Supplementary Figure 7 and adding the following sentences in the Material and Method and Results sections.

Material and Method section:

“Genetic alterations in MPM TCGA series

Genetic alteration data (copy number alteration and mutation) for *CDKN2A*, *NF2*, *BAP1* and *TP53* were retrieved from cBioPortal, an online portal for accessing data from TCGA project (<http://www.cbioportal.org>).”

Results section:

“Then, we characterized E-comp and S-comp at the genetic level by focusing on the major altered genes (*CDKN2A*, *NF2*, *BAP1* and *TP53*) during mesothelial carcinogenesis using available mutation and copy number alteration (CNA) data from the TCGA series. Significant positive associations were observed between *NF2* and *TP53* genetic alterations and the S-score (Supplementary Fig. 7). Interestingly, higher occurrence of *TP53* mutations was reported in MPM tumors with a sarcomatoid contingent ³.”

Discussion was also modified:

“[...] and *TP53* signaling; this finding was consistent with the higher occurrence of *TP53* mutations in tumors with a sarcomatoid contingent.” was replaced by “[...] and *TP53* signaling; this finding was consistent with the association of *TP53* mutations with S-comp.”

4. I am also unclear as to how the different components are maintained in cell lines, which would be expected to be less heterogeneous than tissues. Commenting on these aspects in the paper would be important.

We thank the reviewer for this interesting remark.

Instinctively, less heterogeneity can be expected in cell lines compared to tissues simply because the culture conditions are different from those of a tissue and can exert selective pressure with a different impact on different cell populations.

As shown below, we do see a molecular gradient in cell lines when applying WISP to cell line transcriptomic profiles (series from de Reynies et al. 2014 and from the GDSC).

This observation supports that the establishment of a cell line in culture does not exclude E-comp or S-comp pure tumor cells from a heterogeneous tumor and that this heterogeneity is maintained in cell culture. Other cell types are also preserved in culture such as cancer stem cells (CSC). Published data support that CSC are still present in MPM cell lines (Blum et al., Stem Cell Reports, 2017 PMID: 28285878).

To address this point, we added this comment in the results section:

“Interestingly, the presence of both components in a given cell line supports that tumor heterogeneity was preserved in cell culture. Other cell types are also preserved in culture such as cancer stem cells (CSC). Published data support that CSC are still present in MPM cell lines²⁴.” (Ref 24 :Blum et al., Stem Cell Reports, 2017 PMID: 28285878).“

Reviewer #3 (Remarks to the Author):

The manuscript “Dissecting heterogeneity in MPM through histo-molecular gradients for clinical applications” by Blum and colleagues claims to shed new light on the intra-tumor heterogeneity which could lead to a reconsideration of MPM molecular classification which may have implications for treatment.

Molecular classification of common tumour-types like breast and lung cancer has revealed subtypes that are more susceptible to specific treatments. MPM is a relatively rare tumor-type with currently limited effective treatment options. The current study adds to the previous gene expression based data sets publically available (and cited in the paper) and the clinico-pathology literature to confirm that the underlying histology of MPM predicts clinical behaviour. Showing that the gene expression profile of MPM tumours reflect the gradient in histologies seen by pathologists. The structured clinical application of this information has always been difficult in part because of limited case numbers and treatment options.

1. There are a number of validated prognostic models based upon physical parameters (performance status, weight loss etc) and simple biochemical and haematological factors that are useful for prognostication and selecting treatment options; it is unlikely that, at present, clinicians could justify an additional molecular characterisation of a tumour to change treatment decisions based upon E-comp or S-comp.

We do agree that we are not at that stage yet. However, we believe that, in a near future, the molecular characteristics of the tumor associated to the physical, biochemical and hematological characteristics of the patient will be considered, especially in the selection of

patient treatment options. Cancer genes mutational status is already taken into account in the choice of patient treatment for several cancers. In the particular context of mesothelioma, which does not exhibit drug targetable mutated genes and shows a complex pattern of chromosomal and genetic alterations, focusing on gene expression is particularly relevant as it reflects the chromosomal and genetic alterations and epigenetic changes that occur in the tumor. Expression-based parameters, such as S-comp and E-comp evaluation, if further validated as relevant for prediction of response to treatment and prognostic, could emerge in clinical routine, as current sequencing technologies (e. g. 3' RNA-seq polyA in FFPE samples) make this possible and affordable (< \$100).

2. The reporting of compounds that may be effective against sarcomatoid mesothelioma is a welcome finding.

We thank the reviewer for this encouraging comment.

3. This study uses the software, WISP, available on github to apply a weighting to a set of genes generating an output of two numbers (1) epithelioid component and (2) the sarcomatoid component. I note that having the software hosted by github comes with its imprimatur (i.e. good quality, stable software) – however there appears to be no associated publication or validation of the software. Could the authors provide more justification for the use of this software package for their study?

We perfectly understand the reviewer's concerns.

With regard to the quality of WISP's results:

To estimate the different proportions of pure populations, WISP optimizes a non-negative decomposition algorithm by quadratic programming, which is an algorithm well adapted to the context of signal deconvolution, already used in other deconvolution methods, and which has shown accurate estimations (see for example the DeconRNAseq method based on this algorithm (Gong et al. 2013, PMID 23428642)

Our package is in its stable form and has been applied to several cancers: prostate cancer (Kamoun et al. 2018, PMID: 29945238), colorectal cancer (ESMO 2018 poster #1989), glioma and lung cancer (article in preparation).

Concerning the choice of the method:

Although many deconvolution methods have been developed in recent years, none of them is dedicated to evaluating the presence of a mixture of predefined histo-molecular tumor subtypes. That is why we have added a first step to our methodology consisting in filtering pure samples for the estimation of the pure centroid profiles. Compared to existing deconvolution methods, WISP offers also several improvements such as a warning notification if an estimation is not reliable according to two criteria, the adjusted R-squared and the P-value of the F-test testing how well the data fits the model. It offers as well visualization functions that are not used in the present study.

This is what makes our approach original and explains why we used it in our context.

As this study is not a methodological article, we did not go into the details of the method.

However, in light of the reviewer's concerns, we have added further methodological information.

“Preliminary selection of pure population profiles consisted of MME samples present in subtype C1A (most extreme epithelioid subtype), all MMS and normal samples available in our series. WISP performs an iterative procedure where it estimates for each presupposed pure sample the proportion of the different contingents and removed the samples that do not

mainly contain their corresponding class (WISP default parameters). WISP method uses all the available genes to find the best markers for each pure population. A first gene filtering is based on the P values of the ANOVA test comparing gene expression between all pure populations (FDR adjusted P value < 0.05) and the area under the curve (AUC) score calculated for each pure population (AUC > 0.8). For each pure population, genes are then ranked according to their expression fold change, and the top best markers is retrieved (we chose a maximum number of 50 genes per class). The resulting deconvolution signature is reported in Supplementary Table 10. WISP second step consists in weight estimation by using a non-negative least squares regression model based on the pure population profiles (centroids calculated on the pure population markers) optimized through a quadratic programming algorithm.”

4. From data presented in Table S10 from the WISP package, the combined value of the two output data sets for each sample has a mean of 0.9045 (95% CI 0.89-92) for the 437 cases. Specifically 67% of the samples analysed are reported as being comprised of genes reflecting 90% tumour content. (However, one sample has a combined final total of 0.0997; ie less than 0.1% of genes reflecting tumour content): (Conversely, I note that 5 samples were removed from the analyses – presumably because their combined value was greater than 1). There is no discussion, or presentation of these issues in the main manuscript.

We thank the reviewer for his/her comment.

The following histogram and boxplots show the distribution of the combined value of E-score and S-score in our series (left) and all series (right).

It is true that 67% of the tumor samples analyzed have 90% tumor content based on WISP estimations, but a large majority of tumor samples (90%) shows a tumor content greater than 75%. Therefore, only 10% of the samples have a lower tumor content. Importantly, these tumor samples with low tumor content based on WISP estimations are found in all the analyzed series.

We added this observation in the results section.

“Of note, 90% of the samples analyzed shows a tumor content based on WISP estimations greater than 75% tumor content, suggesting the presence of few tumor samples with lower tumor content, which are found in all tumor series.”

The sample with 0.0997; i.e. less than 0.1% tumor content based on WISP results is the sample “TCGA.LK.A4O4” from the TCGA series. To better understand this very unexpected low value, since no warning was given by WISP regarding this estimation, we used the methylation data available for this sample and used a tool called infiniumPurify (Qin et al. 2018) to estimate the percentage of tumor cells. Consistently, we obtained an estimation of 0.117%. Furthermore, analysis of the mutation profile using cbiportal (http://www.cbiportal.org/patient?studyId=meso_tcga_pan_can_atlas_2018&caseId=TCGA-

LK-A4O4) show that this tumor sample has no CNA et only one mutation with an allele frequency of 0.09. Altogether, these analyses confirm the low tumor content of the TCGA.LK.A4O4 sample, but we do not think it is useful to add these results to our study.

Concerning the 5 samples removed, we mentioned this point in the Material & Method section:

“WISP gives a warning when weight estimations for a sample are unreliable according to several criteria. Consequently, 3 samples were removed from WISP final results in the Gordon series and 2 in the Lopez series.”

As described in our answer to point 1 (reviewer 3), the warning notification is based on the adjusted R-squared and the P-value of the F-test testing how well the data fits the model. We added this clarification in the previous sentence.

“WISP gives a warning when weight estimations for a sample are unreliable according to the adjusted R-squared and the *P* value of the F-test (WISP default parameters were used). Consequently, 3 samples were removed from WISP final results in the Gordon series and 2 in the Lopez series.”

5. Would the authors like to comment on the effect of the tumour purity of the initial sample and how this would affect analysis; are there assumptions made by the WISP program about multiple cell types in the sample? This is important to note in light of subsequent analysis using the MCP counter package.

For the first part of the reviewer's comment, in order to test the potential effect of the tumor purity of the initial sample on our analysis, we perform a Cox proportional hazards

regression model on the samples with a tumor content greater than 90%. As shown below, we found a high significance of the S-comp proportions which is consistent with the results obtained using all samples with a similar Hazard Ratio.

Variable N	Hazard ratio	p
S-score 273		11.09 (6.10, 20.18) <0.001

We added this result as a supplementary figure 12e and add the following sentence in the result section:

“Significance of the S-score was also observed when performing a cox model restricted to patients showing high tumor content samples (combined E-score and S-score>90%).”

Legends of supplementary Figure 12 have also been completed:

“Forest plots of overall survival hazard ratios (HRs) estimated by a multivariate Cox analysis adjusted for series, in MPM patients showing a tumor content higher than 90% (combined E-score and S-score).”

With regard to the second part of the comment, WISP indeed gives an estimate of the proportions of the different tumor populations and the non-tumor component. As mentioned in our answer to the third comment, a preliminary selection of samples with a pure profile is performed. WISP makes therefore an assumption of multiple cell types based on prior knowledge. However, since WISP first step only keeps pure cell populations, if one of these populations does not have specific markers (i.e. the cell population does not actually exist), it will be removed.

MCP-counter is a complementary approach that focuses only on the presence of immune and stromal populations in the tumor sample based on the expression of specific markers of each cell type.

5. From a minor technical point-of-view is it appropriate to include the 8 epithelioid and 3 sarcomatoid samples used to train the model (step 1) in the estimation (step 2)?

These samples are considered pure by WISP and are used in the non-negative least squares regression model to estimate the mixtures in the other samples. However, this should not be considered as a prediction model where these samples should be removed from further analyses (i.e. removed from the test set). What is important in the following step is to look at the estimation of weights in all mixed samples as well as the extreme pure ones in order to characterize the molecular gradient.

6. Could the methods (page 27) be clarified, as I understand 152 genes (Table S9) were in the original deconvolution signature, however it is not clear to me what the 68 genes were “based on a prior selection”. (Similarly, were the qRT-PCR gene lists determined before or after the WISP analysis? as written it appears that the genes used to validate the WISP defined gene set were already known “...WISP method defined a 55 genes signature based on qRT-PCR dataset (Table S1) – this needs to be clarified).

We are sorry we were not clear enough concerning the qRT-PCR deconvolution signature establishment. The 150 genes signature corresponds to the deconvolution signature constructed from frozen tissue transcriptomic profiles only. WISP used for this all the available genes on the Affymetrix array to select the 50 best markers for each component (E-comp, S-comp and the non-tumor component). The 55 genes signature is the deconvolution signature based on qRT-PCR data. These 55 markers were selected by WISP from a qRT-PCR dataset restricted to a set of 68 genes of interest not necessarily in the 150

genes signature that were specific to frozen tissues. The set of 68 genes is described in Supplementary Table 1. These genes were selected based on the correlation with the E-score and S-score in tissue and in cell lines. We added as well genes well-known for being associated with MPM such as *BAP1*, *CDKN2A* and *KRT6A*. For information, these have not been kept by WISP. As shown in Supplementary Fig. 5a and b, the weights estimated from the qRT-PCR data using this 55 genes signature are very strongly correlated to those estimated from the transcriptome microarray data (Affymetrix) using the transcriptomic-based 150 genes signature. This result supports the relevance of the 55 genes signature.

We have modified the text accordingly to provide more clarity on this prior gene selection for qRT-PCR data.

“We generated qRT-PCR data on a prior selection of 68 genes being highly correlated to the E-score and S-score estimated from the transcriptome in tissues and/or cell lines or known in the literature for being associated with MPM. Based on this qRT-PCR dataset, WISP method defined a 55 genes signature (Supplementary Table 1).”

Please also refer to our answer to point 1c. from reviewer 1.

7. Page 27 - the separate analysis on the cell line data to construct “a more appropriate” signature needs to be better justified. I assume this is to reflect the relative purity of cell line samples versus tumour tissue – which ties in with my previous comment.

Indeed, the main difference between cell lines and tissues is the presence of a complex microenvironment in tissues. Therefore, the non-tumor component based on tissue expression profiles is not well suited for cell lines. Although early cell culture passages of the cell lines were used for molecular profiling, cell lines are known to evolve in culture (Gupta PB. et al. Cell. 2011, PMID 21854987, Ben-David, U. et al. Nature 2018, PMID 30089904)

This is why, the best markers for Epithelioid like and Sarcomatoid like components defined in tissue samples are not necessarily the best in cell lines. For all these reasons, we build a deconvolution signature based specifically on cell line expression for applications on expression data from cell lines.

We have added our justification in the text:

“We have built another MPM deconvolution signature dedicated to cell lines. Indeed, the main difference between cell lines and tissues is the presence of a complex microenvironment in tissues. Therefore, the non-tumor component based on tissue expression profiles is not well adapted to cell lines. In addition, the best markers for E-comp and S-comp defined in tissue samples are not necessarily the best in cell lines. Therefore, we used specifically for this signature the transcriptomic data of the cell lines from our previous publication ⁶ and applied the same approach as for tissue samples.”

8. Minor point – Results Page 3 – Supp Fig 1 does not refer to prognosis but rather Supp Fig 2 – which is not referred to in the text.

The reviewer is right. Supp Fig 2 was mentioned, but not at the correct position in the text. Consequently, we moved the reference to Supp Fig 2 in the text to the correct position.

9. For Fig 1b – is it correct that data is present across the whole sample set of 442 samples for the subsets rows CIT; Bueno and TCGA.

Subtypes for the different classification systems (CIT, TCGA and Bueno) were indeed predicted in the whole sample set of 442 samples.

For a better understanding, we clarified this point in the text and in the legend of the figure 1.

Results section:

“These molecular gradients were related to the histology and the different molecular classifications predicted in the whole sample set of 442 samples (Fig. 1b, c, d, Supplementary Fig. 4a, b).”

Legend of the figure 1:

a Correlation matrix of centroid profiles of all clusters from the different classifications. b Estimation of the E-score and S-score and classification subtype predictions in all available tumor tissue samples (442 samples). The samples were ordered based on their E-score and S-score ratios. c-e Boxplots of the E-score and S-score according to the histology results (c), CIT subtype predictions (d), Bueno subtype predictions (e), and TCGA subtype predictions (f). Significance in the T-test comparing E-score and S-score in each modality is shown (*P-value<0.05, **P-value<10e-9, NS: not significant).

For information, the prediction method we used is explained in the Material and Method section:

“*Classifiers* were built for the CIT, Bueno and TCGA classifications based on the calculated centroids. Gene expressions of tumor samples were then correlated (Pearson’s correlation) to all their centroids of a classification system (using only the subset of genes available in both datasets) and the closest centroid class with the highest correlation coefficient was assigned.”

10. From a perspective of readability, the authors may consider for a publication which has the “Results” section before the “Methods” a bit more description in the results text on how analysis was done and on what samples.

We added several methodological precisions throughout the results section underlined in the following sentences:

“First, we performed unsupervised hierarchical clustering, using a consensus method based on 3 different linkages (ward, complete, average) and bootstrap resampling, on our transcriptomic dataset and identified two molecular subtypes of MPM (Supplementary Fig. 1a).”

“We performed a meta-analysis to compare all of the clusters from the different classifications by correlating the centroids of their corresponding meta-profiles (Fig. 1a).”

“We used WISP (<https://cit-bioinfo.github.io/WISP/>, see Method), which is a novel deconvolution method dedicated to intra-tumor heterogeneity assessments, to estimate the proportions of the epithelioid-like and sarcomatoid-like molecular components after preselection of pure population profiles.”

“Then, we performed enrichment analysis of genes whose expression levels correlated with E-score and/or S-score using Kegg, GO and Reactome databases.”

“In addition, we integrated our miRNA dataset, obtained from 60 MPM included in our Affymetrix dataset (CIT exploration series), and identified crucial miRNA-target regulation based on two validated miRNA-target association databases (miRTarbase and TarBase).”

“Interestingly, the presence of S-comp was associated with a worse outcome in each series (Fig. 3a) or when all series were analyzed together (Fig. 3b) using a robust cut-off of 22% of S-score determined after a bootstrap procedure testing thresholds ranging from 0.1 to 0.5 (Supplementary Fig. 11), even when the series were restricted to histologically diagnosed epithelioid MPM (Fig. 3b, Supplementary Fig. 10a, b).”

“A deconvolution signature defined using the transcriptomic data of the cell lines from our previous publication ⁶ was first used to estimate E-score and S-score in GDSC cell lines.”

“In this context, we analyzed immune populations using the Microenvironment Cell Population Counter (MCP-counter) tool on our transcriptomic dataset ²⁵.”

“We also investigated the expression of immune checkpoints using our transcriptomic dataset (Fig. 5b, Supplementary Table 8.A).”

Reviewers' comments:

Reviewer #1 (Remarks to the Author):

This revised manuscript is improved. There are still issues:

1. 3rd line results: "our transcriptomic dataset". Must define this at this point, where from, how many, how done - array vs. RNA-Seq! This issue was noted in the previous review!
2. "from the Bueno series" - no citation!! Give the citation every time you cite the data!!
3. "method dedicated to intra-tumor heterogeneity assessments" weird term to use, dedicated, for a computational method.
4. Although there is a little more detail, there is not enough explanation in the results for the WISP method of analysis or the input data. Presumably this is the RNA data set mentioned above which is not described. In addition, where did the pure population profiles that must be available for WISP come from?
5. State that $E\text{-score} + S\text{-score} = 1$ for each mesothelioma samples.
6. " As expected, 110 genes identified in other studies as overexpressed in MME or in MMS, including UPK3B, MSLN, CLDN15, LOXL2 and VIM, were significantly positively correlated with the E-score or S-score 3, 8. " The direction of the association, e.g. high UPK3B with MME or MMS, should be indicated for each.
7. Gene names in S Fig 6a can barely be read at 500x; for 6b cannot be read at any magnification.
8. p.7 "methylome data and at the miRNA level using miRNome data" - now at least you tell us how many samples, but still give no information on methodology of assessment of methylation and miRNA levels, how many genes covered, what platform.
9. "Several deregulated pathways showed a high proportion of ..." Why do you call them deregulated? I would use the term up-regulated or over-expressed in epithelioid v sarcomatoid mesothelioma.
10. "For instance, the tumor suppressor genes WT1 and PI3KR1 or RUNX1 and PBRM1 were hypermethylated and underexpressed in the high S-comp tumors or high E-comp tumors, respectively. " This sentence construction is complex and not clear.

Reviewer #2 (Remarks to the Author):

The authors have added IHC validation data to the paper, performed additional control and discovery (mutations) analyses I requested earlier. At this stage, I have no further comments - the manuscript is suitable for publication in my view.

Point-by-point response to the referees' comments

First of all, we would like to thank Reviewer 1 for his/her valuable inputs that allowed us to improve our manuscript and Reviewer 2 for his/her positive comments. Please find below a point-by-point response to each of the comments.

Reviewer #1

This revised manuscript is improved. There are still issues:

1. 3rd line results: "our transcriptomic dataset". Must define this at this point, where from, how many, how done - array vs. RNA-Seq! This issue was noted in the previous review!

We are sorry for that and modified this sentence as follows :

"[...] using a consensus method based on 3 different linkages (ward, complete, average) and bootstrap resampling, on the transcriptomic dataset we generated (Affymetrix array on 63 frozen MPM tumor samples) [...]"

We already gave all the details regarding the number of samples and the acquisition of the transcriptomic data in the Materials and Methods, in the sections "Patients" and "mRNA profiling and analysis", respectively.

2. "from the Bueno series" - no citation!! Give the citation every time you cite the data!!

As requested by the reviewer, we have added the references in the following sentences:

"which contained our C1A subtype and the *Epithelioid* subtype from the Bueno series³ or our C2B subtype and the *Sarcomatoid* subtype from the Bueno series³ respectively"

and

"Multivariate Cox analysis integrating the CIT, Gordon, TCGA and Bueno series^{3,6,9}"

3. "method dedicated to intra-tumor heterogeneity assessments" weird term to use, dedicated, for a computational method.

We are sorry for the misuse of the word “dedicated”. We modified the corresponding sentence accordingly both in the result and method sections:

“method aiming at assessing the intra-tumor heterogeneity”

4. Although there is a little more detail, there is not enough explanation in the results for the WISP method of analysis or the input data. Presumably this is the RNA data set mentioned above which is not described. In addition, where did the pure population profiles that must be available for WiSP come from?

We thank the reviewer for this question and would like to clarify what we mean by "pure" population. We consider a MPM sample as a mixture in various proportion of epithelioid-like, sarcomatoid-like and non-tumor component. We call “pure” populations these three components. To identify their specific markers we need pure samples for each. As we collected bulk MPM samples, without sorting any cell populations, we selected representative samples for each component, as it was already described in the “Method section”: “Preliminary selection of pure population profiles consisted in our series of MME samples present in subtype C1A (most extreme epithelioid subtype) for the epithelioid component, all MMS for the sarcomatoid component and available normal samples for the non-tumor component. [...]”

We also remind here that WISP first-step will remove misleading non-pure samples using an iterative procedure from this preselection, as described in the Methods section. In our previous version of the manuscript, we gave a detailed explanation of the WISP method in the method section (“Deconvolution approach” section), and only brief information in the results section. In order to take into account the reviewer's comment, we added in this revised version explanations of the methodology and input data in the results section:

“We used WISP, a novel deconvolution method aiming at assessing intra-tumor heterogeneity by estimating the proportion of pure entities in bulk molecular profiles. WISP is a 2-step approach that first estimates pure population profiles based on predefined pure samples and then estimates the proportion of these pure populations in a mixed sample based on the first step output (see Methods section for more details). We considered a MPM sample as a mixture in various proportion of epithelioid-like, sarcomatoid-like and non-tumor component. For a given sample the sum of these three proportions was therefore equal to 1. The first step of WISP was performed on representative samples for each component from our transcriptomic data (Affymetrix array on 63 tumor samples and 4 normal samples) that were selected as described in the Method section. The second step aiming at estimating the proportion of each component was applied on all available tumor tissue samples (n=442) from our transcriptomic data and the different public transcriptomic datasets (Reynies ⁶, Gordon ⁹, Lopez ⁸, TCGA and Bueno ³ series). We named the epithelioid-like and sarcomatoid-like components, E-comp and S-comp respectively, to distinguish them from the epithelioid and sarcomatoid histologically defined morphologies and their proportions in a given sample the E-score and S-score, respectively.

As shown in Fig. 1b, the E-score and S-score estimated in all available tumor tissue samples (n=442) from the different transcriptomic datasets led to opposite gradients for E-comp and S-comp (Fig. 1b, Supplementary Fig. 4a).”

We also modified the paragraph concerning the predefined selection of pure samples in the method section to provide even more clarity:

“We considered a MPM sample as a mixture in various proportion of epithelioid-like, sarcomatoid-like and non-tumor components. The first step of WISP was performed on a selection of samples from our transcriptomic data (Affymetrix array on 63 tumor samples and 4 normal samples) representative of each component to generate pure population profiles defined on their specific

markers (deconvolution signature). Preliminary selection of pure population profiles consisted in our series of MME samples present in subtype C1A (most extreme epithelioid subtype) for the epithelioid component, all MMS for the sarcomatoid component and available normal samples for the non-tumor component. [...]"

5. State that $E\text{-score} + S\text{-score} = 1$ for each mesothelioma samples.

As already described in the method section, we considered a MPM sample as a mixture in various proportion of an epithelioid-like, a sarcomatoid-like and a non-tumor components. This sentence is now also added in the results section (see response to point 4) and, as suggested by the reviewer, we have clarified that "For a given sample the sum of these three proportions is therefore equal to 1." in the Results section.

6. " As expected, 110 genes identified in other studies as overexpressed in MME or in MMS, including UPK3B, MSLN, CLDN15, LOXL2 and VIM, were significantly positively correlated with the E-score or S-score 3, 8. " The direction of the association, e.g. high UPK3B with MME or MMS, should be indicated for each.

We modified the sentence accordingly.

"As expected, 110 genes identified in other studies as overexpressed in MME or in MMS, were significantly positively correlated with the E-score or S-score. In particular UPK3B, MSLN, CLDN15 were significantly positively correlated to the E-score and LOXL2 and VIM to the S-score."

7. Gene names in S Fig 6a can barely be read at 500x; for 6b cannot be read at any magnification.

We are sorry the gene labels were too small as they are numerous. As the gene names for each pathway are given in Supplementary Table 3A and B for the Supplementary Fig. 6a and 6b respectively, we removed them from the heatmaps and added the following sentence in the legend of Supplementary Fig. 6b.

“The names of the genes associated to the different pathways presented in (a) and (b) are given in Supplementary Table 3A and 3B respectively.”

8. p.7 “methylome data and at the miRNA level using miRNome data” - now at least you tell us how many samples, but still give no information on methodology of assessment of methylation and miRNA levels, how many genes covered, what platform.

We added these clarifications in the Results section:

“Next, we characterized E-comp and S-comp at the epigenetic level using methylome data (HumanMethylation450 Beadchip, 295009 CpG after preprocessing) and at the miRNA level using miRNome data (Illumina HiSeq 2000, 861 miRNA after preprocessing). [...]”

Details on the acquisition of these data were already given in the Materials and Methods in the sections "DNA methylation profiling and analysis" and "microRNA profiling and analysis" of the revised manuscript. The use of Illumina HiSeq 2000 has been specified in the Materials and Methods section by modifying the following sentence:

“The Illumina HiSeq 2000 sequencing platform generated the sequencing images”.

9. “Several deregulated pathways showed a high proportion of ...” Why do you call them deregulated? I would use the term up-regulated or over-expressed in epithelioid v sarcomatoid mesothelioma.

The genes of these pathways are indeed over- or under-expressed depending on the proportion of E-comp or S-comp. This may reflect an activation or inhibition of these pathways along the molecular components. We agree with the reviewer that the term "deregulation" often refers to a situation where there is a comparison with a control condition. To avoid any confusion, we replace the expression "deregulated pathways" by "component-specific pathways", as these pathways are highly associated to E-comp and S-comp (top pathways of the pathway enrichment analyses of genes significantly more expressed along S-comp or E-comp). We modified two sentences of the

result section and two sentences in the discussion, as well as the title and legend of Figure 2 and Supplementary Figure 6 as follows:

Results:

“Several component-specific pathways showed a high proportion of ...”

“Component-specific pathways and epigenetic regulation highlight major actors associated with MPM molecular gradients”

Discussion:

“The methylome data suggest strong regulation of several component-specific pathways”

“miRNA-target networks highlighted candidate master regulators targeting several of the component-specific pathways.”

Figure 2:

“Figure 2 Component-specific pathways and epigenetic regulation.

a Expression heatmap of the component-specific pathways [...] (f) and their targeted pathways (among the component-specific pathways between [...] (e) out of 5 (f) different component-specific pathways [...])”

Supplementary Figure 6:

Component-specific pathways associated to E-comp and S-comp.

Expression heatmaps at the gene level of the component-specific pathways (P value<0.05, Fisher’s exact test) activated along E-comp (a) or S-comp (b).

10. “For instance, the tumor suppressor genes WT1 and PI3KR1 or RUNX1 and PBRM1 were hypermethylated and underexpressed in the high S-comp tumors or high E-comp tumors, respectively. “ This sentence construction is complex and not clear.

We thank the reviewer for his/her comment. We clarified the sentence:

“For instance, the tumor suppressor genes *WT1* and *PI3KR1* were hypermethylated and underexpressed in high E-score tumors whereas *RUNX1* and *PBRM1* were hypermethylated and underexpressed in high S-score tumors.”

Reviewer #2

The authors have added IHC validation data to the paper, performed additional control and discovery (mutations) analyses I requested earlier. At this stage, I have no further comments - the manuscript is suitable for publication in my view.

We thank the Reviewer for his/her positive comments.

REVIEWERS' COMMENTS:

Reviewer #1 (Remarks to the Author):

The authors have responded to my concerns. I have no further issues.